# Trajectory PHD Filter for Adaptive Measurement Noise Covariance Based on Variational Bayesian Approximation

**Xingchen Lu** [1], **Dahai Jing** [1], **Defu Jiang** [1,*], **Yiyue Gao** [2], **Jialin Yang** [1], **Yao Li** [1], **Wendong Li** [1], **Jin Tao** [1] **and Ming Liu** [3]

1  Laboratory of Array and Information Processing, College of Computer and Information, Hohai University, Nanjing 210098, China; lxc10272017@126.com (X.L.); jingdh@hhu.edu.cn (D.J.); karin_yang@hhu.edu.cn (J.Y.); liyaohhu@163.com (Y.L.); 18262629723@163.com (W.L.); taojin@hhu.edu.cn (J.T.)
2  College of Energy and Electrical Engineering, Hohai University, Nanjing 210098, China; gyiyue@hhu.edu.cn
3  The 28th Research Institute of China Electronics Technology Group Corporation, Nanjing 210007, China; liumingpro@gmail.com
*  Correspondence: surfer_jiangdf0801@163.com

**Abstract:** In order to solve the problem that the measurement noise covariance may be unknown or change with time in actual multi-target tracking, this paper brings the variational Bayesian approximation method into the trajectory probability hypothesis density (TPHD) filter and proposes a variational Bayesian TPHD (VB-TPHD) filter to obtain measurement noise covariance adaptively. By modeling the unknown covariance as the random matrix that obeys the inverse gamma distribution, VB-TPHD filter minimizes the Kullback–Leibler divergence (KLD) and estimates the sequence of multi-trajectory states with noise covariance matrices simultaneously. We propose the Gaussian mixture VB-TPHD (AGM-VB-TPHD) filter under adaptive newborn intensity for linear Gaussian models and also give the extended Kalman (AEK-VB-TPHD) filter and unscented Kalman (AUK-VB-TPHD) filter in nonlinear Gaussian models. The simulation results prove the effectiveness of the idea that the VB-TPHD filter can form robust and stable trajectory filtering while learning adaptive measurement noise statistics. Compared with the tag-VB-PHD filter, the estimated error of the VB-TPHD filter is greatly reduced, and the estimation of the trajectory number is more accurate.

**Keywords:** trajectory PHD filter; variational Bayesian approximation; noise covariance matrix; inverse Gamma distribution; estimation of trajectory

## 1. Introduction

Multi-target tracking technology based on random finite sets (RFS) has achieved great development in recent years [1–4]. It not only overcomes the problem of excessive association calculation due to increasing number of targets in traditional multi-target tracking technology based on the data association algorithm [5,6]. It can also solve the tracking difficulties caused by uncertain factors, such as detection parameters and new targets in the environment with an unknown clutter rate [7–9]. The filtering technology developed by RFS has been used in many successful applications, such as aerial warning [10], marine monitoring [11,12], computer vision [13] and sonar detection [14].

PHD filter [15,16] is the first-moment approximation of multi-objective Bayesian filter based on random finite set, which recurses multi-objective density in the single-objective state space. Compared with the cardinalized PHD filter [17], generalized label multi-Bernoulli (GLMB) filter [18], Poisson multi-Bernoulli mixture (PMBM) filter [19] and other filters, the PHD filter is the most basic RFS filter with the lowest computational complexity. The multi-objective posterior density and cardinality propagated by the PHD filter are subjected to Poisson distribution, and the elements in each cardinality are independently and identically distributed.

Take the frame structure of PHD filter into consideration, the dynamic track of the target cannot be obtained directly, and dynamic correlation cannot be automatically generated at different moments of the same target. The key to improve multi-target tracking technology is producing a PHD filter that can form the trajectory without discarding the historical state while capturing the current target state. Instead of labeling the PHD components for trajectory construction [20–22], Angel et al. proposed the TPHD filter for trajectory [23], which uses the trajectory instead of target as the basic variable of trajectory space [24] under the similar structural framework to PHD filter. The trajectory state is estimated in a principled way, which requires a minimum KLD [25] to propagate an approximate PHD subject to the Poisson distribution given by a fast implementation of the Gaussian mixture. On the basis of TPHD, the jump Markov model (JMS) was recently used to achieve highly dynamic maneuvering target's trajectory, and the derivation of JMS-TPHD closed solution proved its tracking effectiveness [26]. In addition, in order to achieve the combination of target tracking and classification, trajectory information was used in [27], combined with the JTC model to classify the trajectory generated by different targets, and the trajectory showed better performance than the PHD filter after confirming the dynamic model based on posterior probability.

It is unlikely to regard the covariance of the measurement noise of the target as a priori knowledge in the practical application, so the measurement noise will change with time followed by the environmental interference. When the estimated noise covariance is different from the actual covariance, PHD will have a partial estimation, resulting in inaccurate state estimation, false target error, wrong cardinality of targets along with the reduced performance and reliability of the filter. To solve the problem of unknown covariance of measurement noise, the least square method was initially used to approximate the noise error [28]. In addition, the interactive multi-model [29] and particle method [30] were also used to solve the problem of measurement noise covariance, but it required a huge amount of calculation. On the other hand, a robust filter with the minimum interference can be achieved under the worst influence of unknown measurement noise on the estimation error [31]. This kind of filter can tolerate the change of noise to the maximum extent rather than realizing adaptive estimation. Refs. [32–34] modeled a random matrix according to the conjugate prior distribution of the covariance. On this basis, variational Bayesian inference is carried out to achieve adaptive noise covariance estimation. The method is mainly used to recurse the joint posterior distribution of the target state and covariance matrix through decomposition of the fixed form distribution of the factors. In comparison, the framework of variational Bayesian approximation can not only effectively reduce the operation cost, but also achieve the purpose of adaptive joint estimation.

We hope to explore an innovative filter that can perform stably under the condition of unknown covariance of measurement noise. By introducing the variational Bayesian approximation into the framework of the TPHD filter (VB-TPHD), we model a matrix of covariance as a random matrix of inverse Gamma distribution and estimate the noise covariance sequence adaptively with the passage of tracking time by combining the joint distribution of the inverse gamma and Gaussian. The trajectories can also be determined more precisely. VB-TPHD can avoid labeling each Gaussian component and managing the track after filtering [35]. In the specific implementation part of the algorithm, we propose the AGM-VB-TPHD filter to be applied to the linear Gaussian model under the condition of unknown newborn intensity [9]. For the nonlinear Gaussian model [36,37], we also provide two solutions, namely the AEK-VB-TPHD filter and AUK-VB-TPHD filter. Two simulation results show their feasibility. Under the condition of unknown and time-varying covariance of measurement noise, the VB-TPHD filter performs better trajectory tracking than the tag-VB-PHD filter does [20,34], and it also has stronger robustness.

The rest of the paper is organized as follows: Section 2 introduces the background and material on trajectory PHD filters. In Section 3, we give the derivation of equations and theoretical basis of the VB-TPHD filter. In Section 4, the implementation of the VB-TPHD

filter under linear and nonlinear conditions is given in detail. Section 5 simulates the tracking performance of the VB-TPHD filter. Finally, Section 6 draws a conclusion.

## 2. Background

### 2.1. Elements

By the target state space extended to trajectory state space, the trace concentration of the basic variable *X* not only contains the target position and velocity information at current time *k*, but also becomes the collection of position and velocity of information since the previous moments; this also means that the expansion of trajectory state from target formation must be considered under the condition of starting time $t$ ($0 \leq t \leq k$) or less. The trajectory duration step is $i$ ($1 \leq i \leq k - t + 1$). After adding the time factor, we represent a single trajectory as the basic variable $X = (t, x^{1:i})$ [24] and the trajectory set space is defined as

$$T_k = \uplus_{(t,i) \in I_k} \{t\} \times \mathbb{R}^{i \times n_x}, \tag{1}$$

where $\times$ represents the Cartesian product, $\uplus$ represents the disconnected union, and $I$ is the range of the above time factor $(t, i)$. In order to better distinguish the states of the trajectory set, element **X** is used to represent the trajectory set and element *X* is used to represent a single trajectory, so the trajectory set or multiple trajectories can be represented by single trajectory *X* as

$$\mathbf{X}_k = \{X_1, \ldots, X_{N_k}\} \in \mathbf{F}(T_k), \tag{2}$$

where $\mathbf{F}(T_k)$ is a subset of the total finite set space of the trajectory set space $T_k$. Similar to the target point, the probability hypothesis density (PHD) $D_p(X)$ based on the trajectory set can be obtained from the set integral below

$$D_p(X) = \int_{\mathbf{F}(T_k)} p(\mathbf{X})\delta\mathbf{X} = \sum_{N_k=0}^{\infty} \frac{1}{N_k!} \int p(\{X_1, \ldots, X_{N_k}\}) dX_{1:N_k}, \tag{3}$$

where $p(\cdot)$ represents the multi-trajectory density. By integrating PHD on a set $\mathbf{F}(T_k)$, the expected number of trajectories on the set can be obtained.

### 2.2. Variational Bayesian

Suppose a multi-target tracking scenario in a two-dimensional scenario at time *k*; there are $n_k$ target states in the state space $\chi$ and $m_k$ measurements are located in the measurement set Z. Thus, multi-objective states and measurements can be expressed by finite sets [15] as

$$x_k = \{x_1, \ldots, x_{n_k}\} \in \chi, \tag{4}$$

$$\mathbf{Z}_k = \{z_1, \ldots, z_{m_k}\} \in Z \tag{5}$$

Because the sensor can be in different environments at different times and the covariance of measurement noise is unknown, it is necessary to estimate the trajectory state and measurement noise jointly in the process of filtering. The covariance of measurement noise is generally set as $R_k = \{R_{k,1}, \ldots \ldots R_{k,m_k}\} \in R$.

Let us assume that the target and the measurement noise are independent, so the dynamic state of the target and the noise covariance matrix are also independent. The joint prior and posterior densities of the target state and the measurement noise covariance can be given by the transformed Chapman-Kolmogorov equation [32]:

$$p(x_k, R_k|z_{1:k-1}) = \int p(x_k|x_{k-1})p(R_k|R_{k-1})p(x_{k-1}, R_{k-1}|z_{1:k-1})dx_{k-1}dR_{k-1}, \tag{6}$$

$$p(x_k, R_k|z_{1:k}) = \frac{p(z_k|x_k, R_k)p(x_k, R_k|z_{1:k-1})}{\int p(z_k|x_k, R_k)p(x_k, R_k|z_{1:k-1})dx_k dR_k}, \tag{7}$$

where $p(z_k|\boldsymbol{x}_k, R_k)$ represents the likelihood function of target and measurement noise at time $k$. Due to $p(R_k|R_{k-1})$ being unknown, the prior $p(\boldsymbol{x}_k, R_k|z_{1:k-1})$ is difficult to obtain, and thus it is unable to launch a posteriori $p(\boldsymbol{x}_k, R_k|z_{1:k})$. Since $\boldsymbol{x}$ and $R$ are coupled in the likelihood function, the variational Bayesian method [32] is used to decouple the coupling. Since the dynamic state of the target is independent of the noise covariance matrix, a posterior $p(\boldsymbol{x}_k, R_k|z_{1:k})$ can be written as

$$p(\boldsymbol{x}_k, R_k|z_{1:k}) \cong Q_X(\boldsymbol{x}_k)Q_R(R_k) \tag{8}$$

In order to find the posterior target state and noise covariance matrix, respectively, the optimal solution is obtained by minimizing KL divergence and variational approximation.

$$\text{KL}[Q_X(\boldsymbol{x}_k)Q_R(R_k)||p(\boldsymbol{x}_k, R_k|z_{1:k})] = \int Q_X(\boldsymbol{x}_k)Q_R(R_k)log\frac{Q_X(\boldsymbol{x}_k)Q_R(R_k)}{p(\boldsymbol{x}_k, R_k|z_{1:k})}d\boldsymbol{x}_k dR_k \tag{9}$$

*2.3. The TPHD Filter*

Imagine a multi-trajectory scenario. Given the posterior PHD at time $k-1$ in the scenario, some of the trajectories will disappear, the surviving trajectories will join up with new target points, and some independent target points will be generated at time $k$. The purpose of the TPHD filter is to identify the true multi-trajectory state from numerous measurements accurately and in a timely manner. The TPHD filter is the output of optimal Poisson approximation by minimizing KLD criterion through Bayesian filtering framework under the condition that the multi-trajectory density is assumed to be Poisson distribution. Before the standard TPHD filter prediction step, the following assumptions [23] are made:

**Assumption 1.** *Each trajectory evolves independently, so the trajectory set at the next time is the combination of surviving trajectories and the point targets of the new birth. The newborn target density $\gamma(\cdot)$ is subjected to the Poisson distribution.*

**Assumption 2.** *However, each trajectory continues to survive with the probability of $p_{s,k}(\cdot)$ and then obtains a point target with the transition density of $f(\cdot)$.*

**Assumption 3.** *The multi-trajectory density $\pi_{k-1}(\cdot)$ obeys the Poisson distribution.*

Based on the above assumptions, the prediction step of single trajectory PHD at time $k$ can be expressed as follows:

$$D_{\omega_k}(X) = D_{\xi_k}(X) + D_{\gamma_k}(X), \tag{10}$$

where

$$D_{\gamma_k}\left(t, x^{1:i}\right) = D_\gamma(t, x)\delta_k[t], \tag{11}$$

$$D_{\xi_k}\left(t, x^{1:i}\right) = p_{s,k}\left(x^{i-1}\right)f\left(x^i|x^{i-1}\right)D_{\pi_{k-1}}\left(t, x^{1:i-1}\right)\delta_{\mathbb{N}_k}[t], \tag{12}$$

where $\delta[\cdot]$ represents the discrete generalized Kronecker delta variable, $\mathbb{N}_k = \{1, \dots, k\}$ represents the set of the time at which the trajectory is located. State transition density $f(\cdot)$ and survival probability $p_{s,k}(\cdot)$ work together in the $k-1$ time trajectory of posterior PHD $D_{\pi_{k-1}}(\cdot)$ to form a surviving trajectory prior PHD $D_{\xi_k}(\cdot)$. According to [24], when $t + i - 1 \neq k$ at the end time of trajectory, the trajectory disappears, which is not considered; only the alive trajectory is considered in Equation (12).

Before the standard TPHD filter update step, the following assumptions [23] are made:

**Assumption 4.** *The trajectory formed by independent target generates the corresponding observation and the target can be detected with the probability of $p_{D,k}(\cdot)$ or omitted with the probability of $1 - p_{D,k}(\cdot)$.*

**Assumption 5.** *The prior trajectory density $\omega(\cdot)$ after the prediction step and the clutter of the scenario obey Poisson distribution.*

**Assumption 6.** *The measurements come from generated targets and clutter independently.*

Based on the above assumptions, the update step of single trajectory PHD at time $k$ can be expressed as follows:

$$D_{\pi_k}(X) = D_{\omega_k}(X)L_{z_k}(X) = D_{\omega_k}(X) \times$$
$$\left( 1 - p_{D,k}(x^i) + p_{D,k}(x^i) \times \sum_{z \in Z_k} \frac{l_k(z|x^i)}{\lambda_c c + \int p_{D,k}(x^i) l_k(z|x^i) D_{\omega_k}^\tau(x^i) dx^i} \right), \tag{13}$$

where

$$D_{\omega_k}^\tau(x^i) = \sum_{t=1}^k \int D_{\omega_k}\left(t, x^{1:k-t+1}\right) dx^{1:k-t}, \tag{14}$$

where $\lambda_c$ is the clutter rate of the trajectory space at time $k$ and $c$ is the reciprocal of the area of the trajectory space. $L_{z_k}(X)$ indirectly denoted that the TPHD trajectory is associated with the measurement of the pseudolikelihood function and measuring likelihood $l_k(z|x^i)$ actually represents the similarity degree between the filtering trajectory and measurement. The higher the degree of similarity, the larger measuring likelihood grows. $D_{\omega_k}^\tau(x^i)$ is the marginal prior multi-trajectory density obtained by integrating the entire trajectory.

What can be seen from the above equation of the prediction step is the new trajectory to track a continuation of the survival and the confirmation at current time; it will not affect the state trajectory of the past time. Instead, the update step is to use measurement at the current time to adjust the long trajectory within limited period. Tracking the target state in the past plays the role of smoothing, which is the critical difference between the conventional PHD filter.

## 3. Adaptive TPHD Filter with Unknown Measurement Noise

### 3.1. The Extended State Space Model

To solve the trajectory filtering in the background of unknown measurement noise covariance, we extend the noise covariance on the target set at a single time to the noise covariance sequence on the trajectory set and then build an extended state space model. We track in state space $\mathbb{T}_k = \biguplus_{(t,i) \in I_k} \{t\} \times \mathbb{R}^{\mathrm{in}_x} \times \mathbb{U}^i$ until time $k$ can set up on the extended trajectory set $\overline{\mathbf{X}}_k = \{\overline{X}_1, \ldots, \overline{X}_{N_K}\} \in F(\mathbb{T}_k)$, $\overline{\mathbf{X}}$ indicates the extended trajectory state set. Define the state element $\overline{X}$ as measurement factors are introduced

$$\overline{X} = \left(X, R^{1:i}\right) = \left(t, x^{1:i}, R^{1:i}\right) \in T_k, \tag{15}$$

where $X$ is defined in the same way as in the Section 2, and $R^{1:i} \in \mathbb{U}^i$ represents the covariance sequence of measurement noise in the extended trajectory state space, which is also different from the conventional trajectory state space. Similar to the point target, given the extended multi-trajectory density function $p(\cdot)$ and the number of trajectories $N_k$, the probability assumption of $D_p(\overline{X})$ bearing based on the extended state space can be obtained by the set integral below:

$$D_p(\overline{X}) = \int_{T_k} p(\overline{\mathbf{X}}) d\overline{\mathbf{X}} = \sum_{N_k=0}^{\infty} \frac{1}{N_k!} \iint p\left(\{X_1, \ldots, X_{N_k}\} \cup \{R_1^{1:i}, \ldots, R_{N_k}^{1:i}\}\right) dX_{1:N_k} dR_{1:N_k}^{1:i} \tag{16}$$

### 3.2. TPHD Using VB Approximation

The development of robust PHD gradually solves the problem of multi-target tracking in the case of unknown detection contour and unknown clutter rate. Comparatively speaking, the processing of unknown measurement noise covariance is more difficult. TPHD provides an excellent framework for us to employ variational means and can also be used to spread the posterior probability of the extended trajectory state $\overline{X}$.

The multi-trajectory density propagated by TPHD filter in the trajectory space is Poisson distribution, which is no longer established in extended state space from Section 3.1.

VB-TPHD minimizes KL divergence to achieve the best Poisson approximation above all. It is assumed that the dynamic models and the measurement noise covariance are independent of each other, and therefore, $p_{s,k}(\overline{X}) = p_{s,k}(x^{i-1})$, $p_{D,k}(\overline{X}) = p_{D,k}(x^i)$.

**Proposition 1.** *Given a posterior PHD $D_{\pi_{k-1}}(\cdot)$, VB-TPHD can be derived from the following equations:*

$$D_{\omega_k}(\overline{X}) = D_{\xi_k}(\overline{X}) + D_{\gamma_k}(\overline{X}), \tag{17}$$

*where*

$$D_{\gamma_k}\left(t, x^{1:i}, R^{1:i}\right) = D_\gamma\left(t, x^{1:i}, R^{1:i}\right)\delta_k[t], \tag{18}$$

$$D_{\xi_k}\left(t, x^{1:i}, R^{1:i}\right) = p_{s,k}\left(x^{i-1}\right)\cdot f\left(x^i|x^{i-1}\right)g\left(R^i|R^{i-1}\right)D_{\pi_{k-1}}\left(t, x^{1:i-1}, R^{1:i-1}\right)\delta_{N_k}[t]. \tag{19}$$

Similar to the background part, VB-TPHD also only considers the survival trajectory. We put the proof of Proposition 1 in Appendix A. When $t + i - 1 \neq k$ at the end of the trajectory, the disappearing trajectory and unknown covariance sequence are not considered.

**Proposition 2.** *Assume that the TPHD can predict the trajectory density $D_{\omega_k}(\overline{X})$ at time $k$, then the VB-TPHD equation can be given as follows:*

$$\begin{aligned}
D_{\pi_k}(\overline{X}) = {}& D_{\omega_k}(\overline{X})\left(1 - p_{D,k}(x^i)\right) \\
& + p_{D,k}(x^i) \times \sum_{z \in Z_k} \frac{l_k\left(z|x^i, R^i\right)D_{\omega_k}\left(t, x^{1:i}, R^{1:i}\right)}{\lambda_c c + \iint p_{D,k} l_k\left(z|x^i, R^i\right)D^\tau_{\omega_k}\left(x^i, R^i\right)dx^i dR^i},
\end{aligned} \tag{20}$$

*where*

$$D^\tau_{\omega_k}\left(x^i, R^i\right) = \sum_{t=1}^k \iint D_{\omega_k}\left(t, x^{1:k-t+1}, R^{1:k-t+1}\right)dx^{1:k-t}dR^{1:k-t} \tag{21}$$

Since the update measurement is only used contacted with the marginalized trajectory density at time $k$, the marginal prior trajectory density $D^\tau_{\omega_k}\left(x^i, R^i\right)$ and measuring density $l_k\left(z|x^i, R^i\right)$ are adopted to calculate the updated trajectory in the extended state space. Proof of Proposition 2 can be seen in Appendix B. In this way, the bridge between the trajectory estimation and target measurement is established. Because $R$ and $g\left(R^i|R^{i-1}\right)$ are unknown, the measuring density caused by $l_k\left(z|x^i, R^i\right)$ is not available that contributes to the filtering disability being able to proceed. We use the variational method to define the joint density function $D_{D_k}\left(x^{1:i}, R^{1:i}|z\right)$:

$$D_{D_k}\left(x^{1:i}, R^{1:i}|z\right) = l_k\left(z|x^i, R^i\right)D_{\omega_k}\left(t, x^{1:i}, R^{1:i}\right) \tag{22}$$

The posterior trajectory state density and noise covariance matrix are separated, and the joint density function is approximately expressed as

$$D_{D_k}\left(x^{1:i}, R^{1:i}|z\right) \cong D_{X_k}\left(x^{1:i}\right)D_{R_k}\left(R^{1:i}\right) \tag{23}$$

The method of minimizing KL divergence is used as shown below. The smaller the KL divergence, the closer the left and right sides of Equation (23).

$$\begin{aligned}
& \mathrm{KL}\left[D_{X_k}\left(x^{1:i}\right)D_{R_k}\left(R^{1:i}\right)\|D_{D_k}\left(x^{1:i}, R^{1:i}|z\right)\right] \\
& = \int D_{X_k}\left(x^{1:i}\right)D_{R_k}\left(R^{1:i}\right)log\left(\frac{D_{X_k}\left(x^{1:i}\right)D_{R_k}\left(R^{1:i}\right)}{D_{D_k}\left(x^{1:i}, R^{1:i}|z\right)}\right)dx^{1:i}dR^{1:i}
\end{aligned} \tag{24}$$

Considering the $D_{X_k}(x^{1:i})$ and $D_{R_k}(R^{1:i})$ are both coupling, fix $D_{R_k}(R^{1:i})$ to obtain $D_{X_k}(x^{1:i})$ and vice versa.

$$D_{X_k}\left(x^{1:i}\right) \propto \exp\left(\int logD_{D_k}\left(x^{1:i}, R^{1:i}, z_k|z_{1:k-1}\right)D_{R_k}\left(R^{1:i}\right)dR\right),$$

$$D_{R_k}\left(R^{1:i}\right) \propto \exp\left(\int logD_{D_k}\left(x^{1:i}, R^{1:i}, z_k|z_{1:k-1}\right)D_{X_k}\left(x^{1:i}\right)dX\right) \tag{25}$$

VB-TPHD not only contains the TPHD key factors including starting time and duration namely time factors as well as trajectory state factor, but also includes track and the corresponding noise covariance sequence. For the trajectory state and covariance matrix coupled in the likelihood function, the accuracy of the estimated covariance matrix increases as the accuracy of the trajectory state estimation is improved at the same time, which is specified in the next section.

## 4. Analytical Implementation of VB-TPHD
### 4.1. The Implementation of Gaussian Mixture VB-TPHD Filter

In the previous section, we obtained the theoretical derivation that variational Bayesian can be well applied to the TPHD filter. In this section, we hope that it can be applied to the Gaussian model to deduce the sequence of measurement noise covariance. It can be seen that the measuring likelihood $l_k\left(z|x^i, R^i\right)$ in Equation (20) only involves measurement noise covariance at the present; therefore, we let VB-TPHD filter perform adaptive estimation of the measurement noise covariance at the current time. Consider that if the state and measurement noise covariance at each moment in the extended state space are included in the calculation, it will add a lot of calculation burden to filter. Therefore, in order to simplify the filtering mechanism of VB-TPHD and improve the filtering efficiency, we only handle the measurement noise covariance at the current time in the tracking process. Thus, the long covariance sequence of measurement noise is not needed to be considered in every single step. Therefore, VB-TPHD derived in the previous section can be further simplified. A posterior PHD of the trajectory at time $k$ can be expressed as $D_{\pi_k}\left(t, x^{1:i-1}, R^i\right)$. Equation (19) in the prediction step and Equation (21) in the update step can be expressed as follows, respectively:

$$\begin{aligned}D_{\xi_k}\left(t, x^{1:i}, R^i\right) &= p_{s,k}(x^{i-1})f\left(x^i, R^i|x^{i-1}, R^{i-1}\right)D_{\pi_{k-1}}\left(t, x^{1:i-1}, R^{i-1}\right)\delta_{N_k}[t]\\&= p_{s,k}(x^{i-1})f\left(x^i|x^{i-1}\right)g\left(R^i|R^{i-1}\right)D_{\pi_{k-1}}\left(t, x^{1:i-1}, R^{i-1}\right)\delta_{N_k}[t],\end{aligned} \tag{26}$$

$$D_{\omega_k}^{\tau}\left(x^i, R^i\right) = \sum_{t=1}^{k}\int D_{\omega_k}\left(t, x^{1:k-t+1}, R^{k-t+1}\right)dx^{1:k-t} \tag{27}$$

In this section, we deduce the concrete implementation of the Gaussian mixture VB-TPHD filter in the scenario of the linear Gaussian model. According to the parameter definition of multi-trajectory PHD, we make the following assumptions:

**Assumption 7.** *The given target obeys the Gaussian linear dynamic model, and the measurement model is also a Gaussian linear model:*

$$f\left(x^i|x^{i-1}\right) = \mathrm{N}\left(x^i;\ \mathrm{F}x^{i-1}, Q\right), \tag{28}$$

$$h\left(z|x^i\right) = \mathrm{N}\left(z;\ \mathrm{H}x^i, R\right), \tag{29}$$

*where $F \in \mathbb{R}^{n_x \times n_x}$ represents the state transition function of the single-moment target, $H \in \mathbb{R}^{n_z \times n_x}$ represents the measurement transition function of the single-moment target, and $Q \in \mathbb{R}^{n_x \times n_x}$ rep-*

resents the known covariance of process noise. $R \in \mathbb{R}^{n_z \times n_z}$ represents the unknown covariance of measurement noise.

Based on Equations (28) and (29), the two densities on the right side of Equation (23) can be set as specified distributions. Let $D_{X_k}(x^{1:i})$ obey the Gaussian distribution. According to inverse gamma distribution being the conjugate prior distribution of the covariance matrix under Gaussian distribution, $D_{R_k}(R^i)$ is modeled as a random matrix which obeys the inverse Gamma distribution and generally the optimal solution of both is obtained by using the variational Bayesian approximation. Inverse gamma distribution turns out to be the natural approximating distribution to achieve the goal with the simple form. $D_{X_k}(x^{1:i})$ with $D_{R_k}(R^i)$ should be designed as follows:

$$D_{X_k}\left(x^{1:i}\right) = \mathrm{N}\left(x^{1:i}; \hat{m}, \hat{P}\right), \; D_{R_k}\left(R^i\right) = \prod_{j=1}^{d} \mathrm{IG}\left(\left(\sigma_j\right)^2; \alpha_j, \beta_j\right) \tag{30}$$

where $\hat{m}$ and $\hat{P}$ denote the mean and covariance of the Gaussian distribution, $d$ represents the dimension of the measurement noise covariance and $\mathrm{IG}\left(\cdot; \alpha_j, \beta_j\right)$ represents the inverse Gamma distribution with freedom parameter $\alpha_j$ and scale parameter $\beta_j$.

### 4.1.1. Newbirth Driven by Measurements

Due to the uncertainty of newborn target location, the prior known intensity is usually artificial to avoid taking the whole detection area into the calculation of the target intensity [15,16] when the TPHD filter initializes the newborn intensity. However, newborn targets may appear in the coverage range of undefined newborn intensity in practical applications, so the filter may ignore such newborn targets and result in missed detection. The measurement values obtained from each scan were adaptively generated to the newborn intensity and the dependence of prior known intensity was removed.

Based on the measurement set $Z_k = \sum_{j=1}^{J_{\gamma_k}} z_k^j$ at time $k$ (the number of measurements $J_{\gamma_k}$), the newborn intensity $D_{\gamma_k}(\overline{X})$ can be modeled as

$$D_{\gamma_k}(\overline{X}) = \sum_{j=1}^{J_{\gamma_k}} w_{\gamma_k}^j \left( \mathrm{N}\left(X; k, m_{\gamma_k}^j, P_{\gamma_k}^j\right) \prod_{l=1}^{d} \mathrm{IG}\left(\left(\sigma_{\gamma_k}^{l,j}\right)^2; \alpha_{\gamma_k}^{l,j}, \beta_{\gamma_k}^{l,j}\right) \right), \tag{31}$$

$$w_{\gamma_k}^j = \frac{K}{J_{\gamma_k}}, \tag{32}$$

$$m_{\gamma_k}^j = H_k^{-1} z_k^j, \tag{33}$$

$$P_{\gamma_k}^j = H_k^{-1} \mathrm{diag}\left(\left(\sigma_{\gamma_k}^{1:d}\right)^2\right) \left(H_k^{-1}\right)^T, \tag{34}$$

where the new generated trajectory states comply with the normal distribution of mean $m_{\gamma_k}^j$ and covariance $P_{\gamma_k}^j$. Additionally, the corresponding covariance matrices of measurement noise are diagonal matrices. $\alpha_{\gamma_k}^l$ and $\beta_{\gamma_k}^l$ are the degrees of freedom and scale parameters of the inverse gamma distribution, respectively. $H_k$ represents the measurement transition function, and $K$ is the constant. Considering that the measurement set at time $k$ is composed of the measurement set generated by clutter and targets, the weight of the new component is set as a uniform small value. The setting of small weight can reduce the amount of the TPHD filter calculation and the impact of large clutter on true targets as well as the false alarm rate. In order to highlight the performance of the adaptive noise covariance estimation of VB-TPHD, $\alpha_{\gamma_k}^l$ and $\beta_{\gamma_k}^l$ are usually fixed and small values, relatively.

### 4.1.2. The Step of Prediction

Given the expansion of the $k - 1$ time trajectory in space trajectory PHD, $\sum_{j=1}^{J_{k-1}} w_{k-1}^j (\mathrm{N}(t, x^{1:i_{k-1}}; t_{k-1}^j, \hat{m}_{k-1}^j, \hat{P}_{k-1}^j) \prod_{l=1}^d \mathrm{IG}((\sigma_{k-1}^{l,j})^2; \alpha_{k-1}^{l,j}, \beta_{k-1}^{l,j}))$, in which the mean $\hat{m}_{k-1}^j \in \mathbb{R}^{i_{k-1} n_x}$, covariance $\hat{P}_{k-1}^j \in \mathbb{R}^{i_{k-1} n_x \times i_{k-1} n_x}$, $t_{k-1}^j$ represents the start time of the trajectory, $i_{k-1}^j$ represents the duration of the trajectory and $j$ is the serial number of the trajectory.

**Assumption 8.** *Survival probability and detection probability are set as constant values in trajectory filter:*

$$p_{s,k}(\overline{X}) = p_s, \, p_{D,k}(\overline{X}) = p_D \tag{35}$$

**Proposition 3.** *Based on Assumptions 1–3 and Assumptions 7 and 8, there is the Gaussian mixture form of VB-TPHD prediction:*

$$D_{\omega_k}(\overline{X}) = D_{\gamma_k}(\overline{X}) + p_s \sum_{j=1}^{J_{\xi_k}} w_{\xi_k}^j \left( \mathrm{N}\left( X; t_{\xi_k}^j, \hat{m}_{\xi_k}^j, \hat{P}_{\xi_k}^j \right) \prod_{l=1}^d \mathrm{IG}\left( \left( \sigma_{\xi_k}^{l,j} \right)^2; \alpha_{\xi_k}^{l,j}, \beta_{\xi_k}^{l,j} \right) \right), \tag{36}$$

*where*

$$w_{\xi_k}^j = p_s w_{k-1}^j, \tag{37}$$

$$\hat{m}_{\xi_k}^j = \left[ (\hat{m}_{k-1}^j)^T, (\dot{F} \cdot \hat{m}_{k-1}^j)^T \right]^T, \tag{38}$$

$$\hat{P}_{\xi_k}^j = \begin{bmatrix} \hat{P}_{k-1}^j & \hat{P}_{k-1}^j \dot{F}^T \\ \dot{F} \hat{P}_{k-1}^j & \dot{F} \hat{P}_{k-1}^j \dot{F}^T + Q \end{bmatrix}, \tag{39}$$

$$\dot{F} = \left[ 0_{1, i_{k-1}^j - 1}, 1 \right] \otimes F, \tag{40}$$

$$\alpha_{\xi_k}^{l,j} = \rho \cdot \alpha_{k-1}^{l,j}, \, \beta_{\xi_k}^{l,j} = \rho \cdot \beta_{k-1}^{l,j}, \tag{41}$$

*where $t_{\xi_k}^j$ represents the starting time of the extended trajectory state and $\rho$ is the attenuation factor which is considered to be suitable for 0.9–0.95 according to [29].*

The above expression can be mainly predicted based on the surviving extended trajectory state $\overline{X}_{\xi_k}$ bearing at $k - 1$. The noise covariance parameters of the surviving extended tracking state at $k - 1$ should be multiplied by the fading factor $\rho$. The proof of Proposition 3 is arranged in Appendix C. The surviving extended tracking component and the newly formed extended trajectory component should be combined as the integral Gaussian components after the step of prediction, $\hat{m}_{\omega_k}^{1:J_{\xi_k}+J_{\gamma_k}} = \left\{ \hat{m}_{\xi_k}^{1:J_{\xi_k}}, m_{\gamma_k}^{1:J_{\gamma_k}} \right\}$, $\hat{P}_{\omega_k}^{1:J_{\xi_k}+J_{\gamma_k}} = \left\{ \hat{P}_{\xi_k}^{1:J_{\xi_k}}, P_{\gamma_k}^{1:J_{\gamma_k}} \right\}$, $w_{\omega_k}^{1:J_{\xi_k}+J_{\gamma_k}} = \left\{ w_{\xi_k}^{1:J_{\xi_k}}, w_{\gamma_k}^{1:J_{\gamma_k}} \right\}$, $\alpha_{\omega_k}^{1:J_{\xi_k}+J_{\gamma_k}} = \left\{ \alpha_{\xi_k}^{1:J_{\xi_k}}, \alpha_{\gamma_k}^{1:J_{\gamma_k}} \right\}$, $\beta_{\omega_k}^{1:J_{\xi_k}+J_{\gamma_k}} = \left\{ \beta_{\xi_k}^{1:J_{\xi_k}}, \beta_{\gamma_k}^{1:J_{\gamma_k}} \right\}$.

### 4.1.3. The Step of Update

After the prediction step, the update step is carried out under the framework of variational Bayesian. Gaussian mixture VB-TPHD calculates the new information and adjusts the mean according to the measurement value $z_k \in Z_k$ and continuously updates the error covariance to obtain the optimal trajectory state and noise covariance.

**Proposition 4.** *Based on Assumptions 4–8, as PHD $D_{\omega_k}(\overline{X})$ can be predicted as the extended trajectory at time k, the extended trajectory update of PHD $D_{\pi_k}(\overline{X})$ can take the following forms:*

$$D_{\pi_k}(\overline{X}) = (1 - p_D) D_{\omega_k}(\overline{X})$$
$$+ p_D \sum_{z_k \in Z_k} \sum_{j=1}^{J_{\xi_k}+J_{\gamma_k}} \left[ w_{\pi_k}^j \mathrm{N}\left( X | t_{\pi_k}^j, \hat{m}_{\pi_k}^j, \hat{P}_{\pi_k}^j \right) \cdot \prod_{l=1}^d \mathrm{IG}\left( \left( \sigma_{\pi_k}^{l,j} \right)^2; \alpha_{\pi_k}^{l,j}, \beta_{\pi_k}^{l,j} \right) \right], \tag{42}$$

$$\alpha_{\pi_k}^{l,j} = \alpha_{\omega_k}^{l,j} + 0.5, \tag{43}$$

$$\dot{H} = \left[ 0_{1,i_k^j-1}, 1 \right] \otimes H, \tag{44}$$

$$\hat{z}_k^j = \dot{H} \hat{m}_{\pi_k}^j, \tag{45}$$

*To obtain the optimal parameters, it is supposed to adjust and update the components. The loop should be started with the record of the loop number n:*

$$\left( \sigma_{\pi_k}^{l,j,n} \right)^2 = \mathrm{diag}\left( \frac{\beta_{\pi_k}^{1,j,n}}{\alpha_{\pi_k}^{1,j}}, \ldots \ldots, \frac{\beta_{\pi_k}^{d,j,n}}{\alpha_{\pi_k}^{d,j}} \right) \tag{46}$$

$$S_k^{j,n} = \left( \sigma_{\pi_k}^{l,j,n} \right)^2 + \dot{H} \hat{P}_{\omega_k}^j \left( \dot{H} \right)^T, \tag{47}$$

$$K_k^{j,n} = \hat{P}_{\omega_k}^j \left( \dot{H} \right)^T \left( S_k^{j,n} \right)^{-1}, \tag{48}$$

$$\hat{m}_{\pi_k}^{j,n} = \hat{m}_{\omega_k}^j + K_k^{j,n} \left( z - \hat{z}_k^j \right), \tag{49}$$

$$\hat{P}_{\pi_k}^{j,n} = \hat{P}_{\omega_k}^j - K_k^{j,n} \dot{H} \hat{P}_{\omega_k}^j, \tag{50}$$

$$\beta_{\pi_k}^{j,l,n+1} = \beta_{\omega_k}^{j,l} + 0.5 \times \left( z_k - \dot{H} \hat{m}_k^{j,n} \right)^2 + 0.5 \times \left( \dot{H} \hat{P}_{\pi_k}^{j,n} \left( \dot{H} \right)^T \right), \tag{51}$$

*Assume that the predicted track set is updated according to the measurement set at time k and the cycle number n is recorded. Break the loop, and export $\hat{m}_{\pi_k}^{j,n}, \hat{P}_{\pi_k}^{j,n}, S_k^{j,n}$ and $\left( \sigma_{\pi_k}^{l,j,n} \right)^2$ when $\hat{m}_{\pi_k}^{j,n}$ tends to be a fixed value to calculate the weights of each Gaussian component updated:*

$$w_{\pi_k}^j = \frac{p_D w_{\omega_k}^j \mathrm{N}\left( \hat{z}_k^j, S_k^{j,n} \right)}{\lambda_c V + p_D \sum_{j=1}^{J_{\xi_k}+J_{\gamma_k}} w_{\omega_k}^j \mathrm{N}\left( \hat{z}_k^j, S_k^{j,n} \right)} \tag{52}$$

We put the proof of Proposition 4 in Appendix D.

### 4.1.4. Other Contents about the Implementation

The biggest difference between TPHD and PHD lies in the different object within the filter processing, although they have similar prediction and update steps. The former is the processing of state sequence, namely trajectory. The longer the filtering time, the more states need to be calculated and stored as well as the greater the computational burden [23]. Here, the L-scan trajectory processing method is proposed. When the update window length $L$ is small, its computational efficiency is almost the same as PHD. When the filtering duration $I < L$, only the trajectory state within the duration range is processed, while once the filtering duration $I \geq L$, only the trajectory state within the current moment, and the previous $L$ time is processed.

It is necessary to prune and absorb the Gaussian components in the current extended state space after the update step in order to ensure the efficiency of operation and achieve

better joint extraction of state and noise covariance. Set the upper limit of the number of Gaussian components $I_{max}$ and the weight threshold $\varepsilon_p$. Cut out the weight number $I$ whose weight is greater than the threshold:

$$I = \left\{ j \in \{1, \ldots, J_k\}: w_k^j > \varepsilon_p \right\} \tag{53}$$

Above, the number of Gaussian components is directly processed from the threshold value. In order to ensure the validity of the remaining Gaussian components, the current Gaussian components with high similarity are retained based on the Mahalanobis distance principle. This process is absorption. Specifically, by calculating the Mahalanobis distance between each Gaussian component and the maximum Gaussian component of the current moment weight, if the distance is less than the set threshold $\varepsilon_a$, the serial number $S$ of Gaussian component is selected as

$$S = \left\{ j \in I : \left( \hat{m}_k^j \hat{m}_k^i \right)^T \left( \hat{P}_k^j \right)^{-1} \left( \hat{m}_k^j - \hat{m}_k^i \right) \leq \varepsilon_a \right\}, \tag{54}$$

where $i = \operatorname{argmax}(w_k^j)$, then the weight of one of the Gaussian components corresponding to $L$ is appropriately increased and the other Gaussian components are deleted so as to achieve the purpose of controlling the number of Gaussian components.

The last step of VB-TPHD is the extraction of trajectory in extended state space. Similar to PHD, the number of trajectory can be estimated as

$$N_k = round \left( \sum_{j=1}^{J_k} w_k^j \right) \tag{55}$$

Sort all Gaussian components from high to low according to their weights, screen out $N_k$ Gaussian components with high weight and finally extract the joint covariance of trajectory as follows:

$$\left\{ \left( t_1, i_k^j, \hat{m}_k^j, \left( \alpha_k^{l,j}, \beta_k^{l,j} \right)_{l=1}^d \right) \right\}, j = 1 : N_k \tag{56}$$

### 4.2. Nonlinear Implementation of VB-TPHD Filter

The TPHD filter can provide closed solution under iteration, which can be regarded as a very effective multi-trajectory filtering method under the linear Gaussian model and can obtain the ideal effect through the Gaussian mixture form based on the Kalman filter. Now assume that both the state process and the measurement process are nonlinear Gaussian models [16]:

$$x_k = \varphi_k(x_{k-1}, v_k), \tag{57}$$

$$z_k = h_k(x_k, v_k), \tag{58}$$

where $\varphi_k(\cdot)$ and $h_k(\cdot)$ are known nonlinear dynamic and measurement equations, $v_k$ and $v_k$ are state noise and the measurement noise of Gaussian distribution, and $Q_k$ and $R_k$ are covariances of state noise and measurement noise, respectively.

It is not feasible to use the Gaussian mixture VB-TPHD filter to process nonlinear Gaussian models directly, but it can still be applied to nonlinear systems through an improved Kalman filter, such as extended Kalman (EK) filter for linearing locally nonlinear Gaussian systems [36] and an unscented Kalman (UK) filter that approximates nonlinear Gaussian systems using precisely selected $\sigma$ points [37,38]. Then, the realization of AEK-VB-TPHD and AUK-VB-TPHD is given in detail.

The key steps of AEK-VB-TPHD and AUK-VB-TPHD are shown in Algorithms 1 and 2. These two filters not only adopt the L-scan method to avoid including the whole trajectories into the calculation like AGM-VB-TPHD, but also execute the operation of

pruning, absorption and state extraction according to Equations (53)–(56). It is worth mentioning that because TPHD is the trajectory involved in the iterative cycle and the filtering state is multi-dimensional, the weight $u$ of $\sigma$ points will be negative, which leads to the negative determination of the estimated covariance matrix and the next Cholesky decomposition cannot be carried out. Therefore, SVD [38] is used for the decomposition of the matrix in AUK-VB-TPHD.

---

**Algorithm 1.** Prediction and Update for the AEK-VB-TPHD Filter.

---

**Step** 1. (Prediction for newborn targets)

  **Input:** $z_k \in Z_k$, $K$, $h_k^{-1}$.

  **Output:** $\{w_{\gamma_k}^j, m_{\gamma_k}^i, P_{\gamma_k}^i, (\alpha_{\gamma_k}^{l,j}, \beta_{\gamma_k}^{l,j})_{l=1}^d\}_{j=1}^{J_{\gamma_k}}$.

      for $j = 1 : J_{\gamma_k}$ (number of measurements)

        use Equations (32)–(34).

      end for

**Step** 2. (Prediction for existing trajectories)

  **Input:** $p_s$, $\Phi_k^j$, $v_k^j$, $\rho$, $\{w_{\pi_{k-1}}^j, \hat{m}_{\pi_{k-1}}^j, \hat{P}_{\pi_{k-1}}^j, (\alpha_{\pi_{k-1}}^{l,j}, \beta_{\pi_{k-1}}^{l,j})_{l=1}^d\}_{j=1}^{J_{\pi_{k-1}}}$.

  **Output:** $\{w_{\xi_k}^j, \hat{m}_{\xi_k}^j, \hat{P}_{\xi_k}^j, (\alpha_{\xi_k}^{l,j}, \beta_{\xi_k}^{l,j})_{l=1}^d\}_{j=1}^{J_{\xi_k}}$.

      for $j = 1, \ldots, J_{\xi_k}$

        use Equations (37)–(38) to obtain $w_{\xi_k}^j, \hat{m}_{\xi_k}^j$.

$$\hat{P}_{\xi_k}^j = \begin{bmatrix} \hat{P}_{\pi_{k-1}}^j & \hat{P}_{\pi_{k-1}}^j \left(\dot{F}_{k-1}^j\right)^T \\ \dot{F}_{k-1}^j \hat{P}_{\pi_{k-1}}^j & \dot{F}_{k-1}^j \hat{P}_{\pi_{k-1}}^j \left(\dot{F}_{k-1}^j\right)^T + G_{k-1}^j Q_{k-1} \left(G_{k-1}^j\right)^T \end{bmatrix},$$

        Where $\dot{F}_{k-1}^j = \left[0_{1,i_{k-1}^j-1}, 1\right] \otimes F_{k-1}^j$, $F_{k-1}^j = \left.\frac{\partial \Phi_k^j(x_{k-1},0)}{\partial x_{k-1}}\right|_{x_{k-1}=m_{k-1}^j}$,

$$G_{k-1}^j = \left.\frac{\partial \Phi_k^j\left(m_{k-1}^j, v_k^j\right)}{\partial v_k^j}\right|_{v_k^j=0}.$$

        use Equation (41) to obtain $(\alpha_{\xi_k}^{l,j}, \beta_{\xi_k}^{l,j})_{l=1}^d$.

      end for

**Step** 3. (Update for all trajectories)

  **Input:** $p_D$, $l_z = J_{\gamma_k}$, $H_k^j$, $\{w_{\omega_k}^j, \hat{m}_{\omega_k}^j, \hat{P}_{\omega_k}^j, (\alpha_{\omega_k}^{l,j}, \beta_{\omega_k}^{l,j})_{l=1}^d\}_{j=1}^{J_{\omega_k}=J_{\gamma_k}+J_{\xi_k}}$ (combine Output of Step

1&2)

  **Output:** $\{w_{\pi_k}^j, \hat{m}_{\pi_k}^j, \hat{P}_{\pi_k}^j, (\alpha_{\pi_k}^{l,j}, \beta_{\pi_k}^{l,j})_{l=1}^d\}_{j=1}^{J_{\pi_k}=(l_z+1)J_{\omega_k}}$.

      (Update for undetected trajectories)

        for $j = 1, \ldots, J_{\omega_k}$

        $w_{\pi_k}^j = (1 - p_D)w_{\omega_k}^j$,

        $\hat{m}_{\pi_k}^j = \hat{m}_{\omega_k}^j$, $\hat{P}_{\pi_k}^j = \hat{P}_{\omega_k}^j$,

        $\alpha_{\pi_k}^{l,j} = \alpha_{\omega_k}^{l,j}$, $\beta_{\pi_k}^{l,j} = \beta_{\omega_k}^{l,j}$, for $l = 1, \ldots, d$.

        end for

      (Update for detected trajectories)

        for each $z_k \in Z_k$

        for $j = 1, \ldots, J_{\omega_k}$

        $\alpha_{\pi_k}^{l,j} = \alpha_{\omega_k}^{l,j} + 0.5$, $\beta_{\pi_k}^{l,j} = \beta_{\omega_k}^{l,j}$, for $l = 1, \ldots, d$.

        $\hat{m}_{\pi_k}^{j,0} = \hat{m}_{\omega_k}^j$, $\hat{P}_{\pi_k}^{j,0} = \hat{P}_{\omega_k}^j$.

        $H_k^j = \left.\frac{\partial H_k^j(x_k,0)}{\partial x_k}\right|_{x_k=m_{\omega_k}^j}$, $\dot{H}_k^j = \left[0_{1,i_k^j-1}, 1\right] \otimes H_k^j$, $U_k^j = \left.\frac{\partial H_k^j\left(m_{\omega_k}^j, R_{\omega_k}^j\right)}{\partial v_k^j}\right|_{R_{\omega_k}^j=0}$.

        Then iterate the following a few. e.g., $N$, steps:

        use Equations (46) and (48)–(51) to obtain $\hat{m}_{\pi_k}^{j,n}, \hat{P}_{\pi_k}^{j,n}, \beta_{\pi_k}^{l,j,n}, R_{\pi_k}^{l,j,n}$

        where $S_k^{j,n} = U_k^j R_{\pi_k}^{l,j,n} (U_k^j)^T + \dot{H}_k^j \hat{P}_{\omega_k}^j \left(\dot{H}_k^j\right)^T$.

---

---

**Algorithm 1.** *Cont.*

---

$$\text{Set } \hat{m}_{\pi_k}^j = \hat{m}_{\pi_k}^{j,N}, \hat{P}_{\pi_k}^j = \hat{P}_{\pi_k}^{j,N}, \beta_{\pi_k}^{l,j} = \beta_{\pi_k}^{l,j,N}.$$

$$Q^j(z_k) = N\left(z_k; \dot{H}_{\omega_k}^j \hat{m}_k^j, S_k^{j,n}\right), w_{\pi_k}^{l_z J_{\omega_k}+j} = p_D w_{\omega_k}^j Q^j(z_k).$$

end for

$$w_{\pi_k}^{l_z J_{\omega_k}+j} = \frac{w_{\pi_k}^{l_z J_{\omega_k}+j}}{\lambda_c V + \sum_{i=1}^{J_{\omega_k}} w_{\pi_k}^{l_z J_{\omega_k}+i}}, \text{ for } j = 1, \ldots, J_{\omega_k}$$

end for

---

**Algorithm 2.** Prediction and Update for the AUK-VB-TPHD Filter.

---

**Step** 1. (Prediction for newborn targets)

follow Step 1 of **Algorithm 1**.

**Step** 2. (Prediction for existing trajectories)

**Input:** $p_s, \Phi_k^j, v_k^j, \alpha, \beta, k, \rho, \{w_{\pi_{k-1}}^j, \hat{m}_{\pi_{k-1}}^j, \hat{P}_{\pi_{k-1}}^j, (\alpha_{\pi_{k-1}}^{l,j}, \beta_{\pi_{k-1}}^{l,j})_{l=1}^d\}_{j=1}^{J_{\pi_{k-1}}}$.

**Output:** $\{w_{\zeta_k}^j, \hat{m}_{\zeta_k}^j, \hat{P}_{\zeta_k}^j, (\alpha_{\zeta_k}^{l,j}, \beta_{\zeta_k}^{l,j})_{l=1}^d\}_{j=1}^{J_{\zeta_k}}$

for $j = 1, \ldots, J_{\zeta_k}$

$$w_{\zeta_k}^j = p_s w_{\zeta_k}^j, \dot{m}_{\zeta_{k-1}}^j = \begin{bmatrix} m_{\pi_{k-1}}^j \\ 0 \end{bmatrix}, \dot{P}_{\zeta_{k-1}}^j = \begin{bmatrix} P_{\pi_{k-1}}^j & 0 \\ 0 & Q_{k-1} \end{bmatrix}$$

$$\left\{\dot{m}_{\zeta_{k-1}}^{j,s}, u_{\zeta_{k-1}}^{j,s}, s\right\} = \text{ut}\left\{\dot{m}_{\zeta_{k-1}}^j, \dot{P}_{\zeta_{k-1}}^j, \alpha, \beta, k\right\}. \text{ for } s = 0, \ldots, 2NUM.$$

$$\dot{m}_{\zeta_{k-1}}^j = \sum_{s=0}^{2n} u_{\zeta_{k-1}}^{j,s} \Phi_k\left(\dot{m}_{\zeta_{k-1}}^{j,s}, v_k\right),$$

$$\dot{X}_{\zeta_{k-1}}^j = \sum_{s=0}^{2n} u_{\zeta_{k-1}}^{j,s}\left(\dot{m}_{\zeta_{k-1}}^j - \Phi_k^j\left(\dot{m}_{\zeta_{k-1}}^{j,s}, v_k\right)\right),$$

$$\hat{m}_{\zeta_k}^j = \left[\left(\hat{m}_{\pi_{k-1}}^j\right)^T, \left(\dot{m}_{\zeta_{k-1}}^j\right)^T\right]^T, \hat{P}_{\zeta_k}^i = \begin{bmatrix} \hat{P}_{\pi_{k-1}}^j & \hat{P}_{\pi_{k-1}}^j\left(\dot{F}_{k-1}^j\right)^T \\ \dot{F}_{k-1}^j \hat{P}_{\pi_{k-1}}^j & \dot{X}_{\zeta_{k-1}}^j \text{diag}(u_{\zeta_{k-1}}^{j,s})\left(\dot{X}_{\zeta_{k-1}}^j\right)^T \end{bmatrix},$$

where $\dot{F}_{k-1}^j = \left[0_{1,i_{k-1}^j-1}, 1\right] \otimes \Phi_k^j$.

use Equation (41) to obtain $(\alpha_{\zeta_k}^{l,j}, \beta_{\zeta_k}^{l,j})_{l=1}^d$.

end for

**Step** 3. (Update for all trajectories)

**Input:** $p_D, l_z = J_{\gamma_k}, \alpha, \beta, k, h_k, \{w_{\omega_k}^j, \hat{m}_{\omega_k}^j, \hat{P}_{\omega_k}^j, (\alpha_{\omega_k}^{l,j}, \beta_{\omega_k}^{l,j})_{l=1}^d\}_{j=1}^{J_{\omega_k} = J_{\gamma_k} + J_{\zeta_k}}$ (combine Output of Step 1&2)

**Output:** $\{w_{\pi_k}^j, \hat{m}_{\pi_k}^j, \hat{P}_{\pi_k}^j, (\alpha_{\pi_k}^{l,j}, \beta_{\pi_k}^{l,j})_{l=1}^d\}_{j=1}^{J_{\pi_k} = (l_z+1)J_{\omega_k}}$.

(Update for undetected trajectories)

for $j = 1, \ldots, J_{\omega_k}$

$w_{\pi_k}^j = (1 - p_D)w_{\omega_k}^j$,

$\hat{m}_{\pi_k}^j = \hat{m}_{\omega_k}^j, \hat{P}_{\pi_k}^j = \hat{P}_{\omega_k}^j$,

$\alpha_{\pi_k}^{l,j} = \alpha_{\omega_k}^{l,j}, \beta_{\pi_k}^{l,j} = \beta_{\omega_k}^{l,j}$, for $l = 1, \ldots, d$.

end for

(Update for detected trajectories)

for each $z_k \in Z_k$

for $j = 1, \ldots, J_{\omega_k}$

$\alpha_{\pi_k}^{l,j} = \alpha_{\omega_k}^{l,j} + 0.5, \beta_{\pi_k}^{l,j} = \beta_{\omega_k}^{l,j}$, for $l = 1, \ldots, d$.

$\hat{m}_{\pi_k}^{j,0} = \hat{m}_{\omega_k}^j, \hat{P}_{\pi_k}^{j,0} = \hat{P}_{\omega_k}^j$.

Then iterate the following a few. e.g., *N*, steps:

use Equation (46) to obtain $R_{\pi_k}^{l,j,n}$

$$\dot{m}_{\pi_k}^{j,n} = \begin{bmatrix} \hat{m}_{\pi_k}^{j,n} \\ 0 \end{bmatrix}, \dot{P}_{\pi_k}^{j,n} = \begin{bmatrix} \hat{P}_{\pi_k}^{j,n} & 0 \\ 0 & R_{\pi_k}^{l,j,n} \end{bmatrix}$$

---

---

**Algorithm 2.** *Cont.*

$$\left\{ \dot{z}_{\pi_k}^{j,n,s}, u_{\pi_k}^{j,n,s}, s \right\} = \text{ut} \left\{ \dot{m}_{\pi_k}^{j,n}, \dot{P}_{\pi_k}^{j,n}, \alpha, \beta, k \right\}. \text{ for } s = 0, \ldots, 2NUM$$

$$\dot{z}_{\pi_k}^{j,n} = \sum_{s=0}^{2n} u_{\pi_k}^{j,n,s} h_k(\dot{z}_{\pi_k}^{j,n,s}, R_{\pi_k}^{l,j,n}), \quad \dot{Z}_{\pi_k}^{j,n} = \sum_{s=0}^{2n} u_{\pi_k}^{j,n,s} \cdot (\dot{z}_{\pi_k}^{j,n} - h_k(\dot{z}_{\pi_k}^{j,n,s}, R_{\pi_k}^{l,j,n})),$$

$$S_k^{j,n} = \sum_{s=0}^{2n} u_{\pi_k}^{j,n,s} \dot{Z}_{\pi_k}^{j,n} (\dot{Z}_{\pi_k}^{j,n})^T, \quad \dot{G}_k^{j,n} = \sum_{s=0}^{2n} u_{\pi_k}^{j,n,s} \cdot (\dot{z}_{\pi_k}^{j,n} - \dot{z}_{\pi_k}^{j,n,s})(\dot{z}_{\pi_k}^{j,n} - \dot{z}_{\pi_k}^{j,n,s})^T,$$

$$K_k^{j,n} = \dot{G}_k^{j,n} / S_k^{j,n},$$

$$\hat{m}_{\pi_k}^{j,n} = \hat{m}_{\omega_k}^j + K_k^{j,n}\left( z - \hat{z}_k^j \right), \hat{P}_{\pi_k}^{j,n} = \hat{P}_{\omega_k}^j - K_k^{j,n} \times \left( \dot{G}_k^{j,n} \right)^T,$$

$$\beta_{\pi_k}^{l,j,n+1} = \beta_{\omega_k}^{l,j} + 0.5 \times \left( z_k - \dot{H}_k^j \hat{m}_{\pi_k}^{j,n} \right)^2 + 0.5 \times \left( \sum_{s=0}^{2n} u_{\pi_k}^{j,s} \dot{z}_{\pi_k}^j (\dot{z}_{\pi_k}^j)^T \right),$$

Set $\hat{m}_{\pi_k}^j = \hat{m}_{\pi_k}^{j,N}, \hat{P}_{\pi_k}^j = \hat{P}_{\pi_k}^{j,N}, \beta_{\pi_k}^{l,j} = \beta_{\pi_k}^{l,j,N}, S_k^j = S_k^{j,N}, \dot{z}_{\pi_k}^j = \dot{z}_{\pi_k}^{j,1}$

$$Q^j(z_k) = N\left( z_k; \dot{z}_{\omega_k}^j, S_k^j \right), w_{\pi_k}^{l_z J \omega_k + j} = p_D w_{\omega_k}^j Q^j(z_k).$$

end for

$$w_{\pi_k}^{l_z J \omega_k + j} = \frac{w_{\pi_k}^{l_z J \omega_k + j}}{\lambda_c V + \sum_{i=1}^{J_{\omega_k}} w_{\pi_k}^{l_z J \omega_k + i}}, \text{ for } j = 1, \ldots, J_{\omega_k}$$

end for

---

**Remark 1.** *Noted that AUK-VB-TPHD is more suitable to non-differentiable nonlinear models than AEK-VB-TPHD, which avoids the complicated and error-prone Jacobian matrix calculation.*

## 5. Simulation and Discussion

### 5.1. Scenario 1

To prove the tracking performance of Gaussian mixture VB-TPHD, a two-dimensional extended trajectory space scenario with unknown and constant noise covariance is designed in Scenario 1. Suppose there are four targets can form trajectories in the scenario and each target obeys the linear Gaussian dynamic and measurement model such as Equations (28) and (29), that is, it moves at a constant velocity. Each model is expressed as follows:

$$F_k = \begin{bmatrix} I_2 & \Delta I_2 \\ 0_2 & I_2 \end{bmatrix}, \quad Q_k = \sigma_v^2 \begin{bmatrix} \frac{\Delta^4}{4} I_2 & \frac{\Delta^3}{3} I_2 \\ \frac{\Delta^3}{3} I_2 & \Delta^2 I_2 \end{bmatrix}, \tag{59}$$

$$H_k = \begin{bmatrix} I_2 & 0_2 \end{bmatrix}, \quad R_k = \text{diag}(\sigma_{v_1}^2, \sigma_{v_2}^2),$$

where $I_2$ and $0_2$ represent $2 \times 2$ identity matrix and zero matrix, respectively, and $\Delta$ represents the scanning period. In scenario 1, $\Delta = 1$ s, $\sigma_v = 1.8$ m/s represents the standard deviation of the process noise. Similarly, $\sigma_{v_1}$ and $\sigma_{v_2}$ represent the unknown quantity of the standard deviation of the measurement noise. The state of the target in the extended space can be written as follows: the first and second dimensions are the $x$ and $y$ coordinates of the target, respectively, and the third and fourth dimensions are the fractional velocities of the target in the $x$ and $y$ direction respectively. The first and second dimensions of measurement obtained are the $x$ and $y$ coordinates of the measurement, respectively. Next are some settings for the scenario parameters. The size of scenario is set to [0, 1000] m × [0, 1000] m and the total tracing duration is set to 100 s. The trajectory survival probability is set as $p_s = 0.99$ and the detection probability is set as $p_D = 0.98$. The clutter generated in each scan obeys the uniform distribution and its number obeys the Poisson distribution with the parameter $\lambda = 5$. The true unknown measurement noise covariance $R_k$ is set to be diag(4, 4).

The initial state, birth time and death time of the trajectory in the specific extended state space are shown in Table 1. The threshold of pruning is set as $\varepsilon_p = 10^{-4}$, the threshold of absorption is set as $\varepsilon_a = 4$ and the maximum Gaussian component number is set as $J_{\max} = 70$. The simulation in scenario 1 adopts L-scan approximation whose $L = 5$. The covariance matrix for measurement noise obeys the inverse Gamma distribution. The initial parameters are $\alpha_1 = \beta_1 = 1$, $\alpha_2 = \beta_2 = 1$, and the given fading factor is $\rho = 0.95$.

**Table 1.** The information of true targets.

|  | Initial Targets States | Birth Time/s | Death Time/s |
|---|---|---|---|
| Target 1 | {100; 400; 12.14; 4.86} | 2 | 45 |
| Target 2 | {450; 300; −3.43; 4.29} | 2 | 65 |
| Target 3 | {100; 150; 6.15; 0.77} | 15 | 85 |
| Target 4 | {100; 400; 3.75; −3.75} | 20 | 100 |

In scenario 1, we conduct 1000 Monte Carlo runs with MATLAB 2021a on 2.50 GHz Intel I5 laptop. GOSPA metric [39] is used to evaluate the performance of extended trajectory estimation. Simulation parameters are set as $p = 2$, $c = 10$, $\alpha = 2$. The GOSPA error of the target is decomposed into GOSPA localization cost, GOSPA missed target cost and GOSPA false target cost. From these aspects, the root mean square error (RMS) of Monte Carlo runs is obtained to analyze the filter.

Figure 1 reflects the true tracking of the four targets and it can be seen that a good track is formed in the extended state space. Figures 2–4 respectively show the average standard deviation of the measurement noise, the average cardinality estimated by AGM-VB-TPHD and the comparison with the other filters under the GOSPA metric.

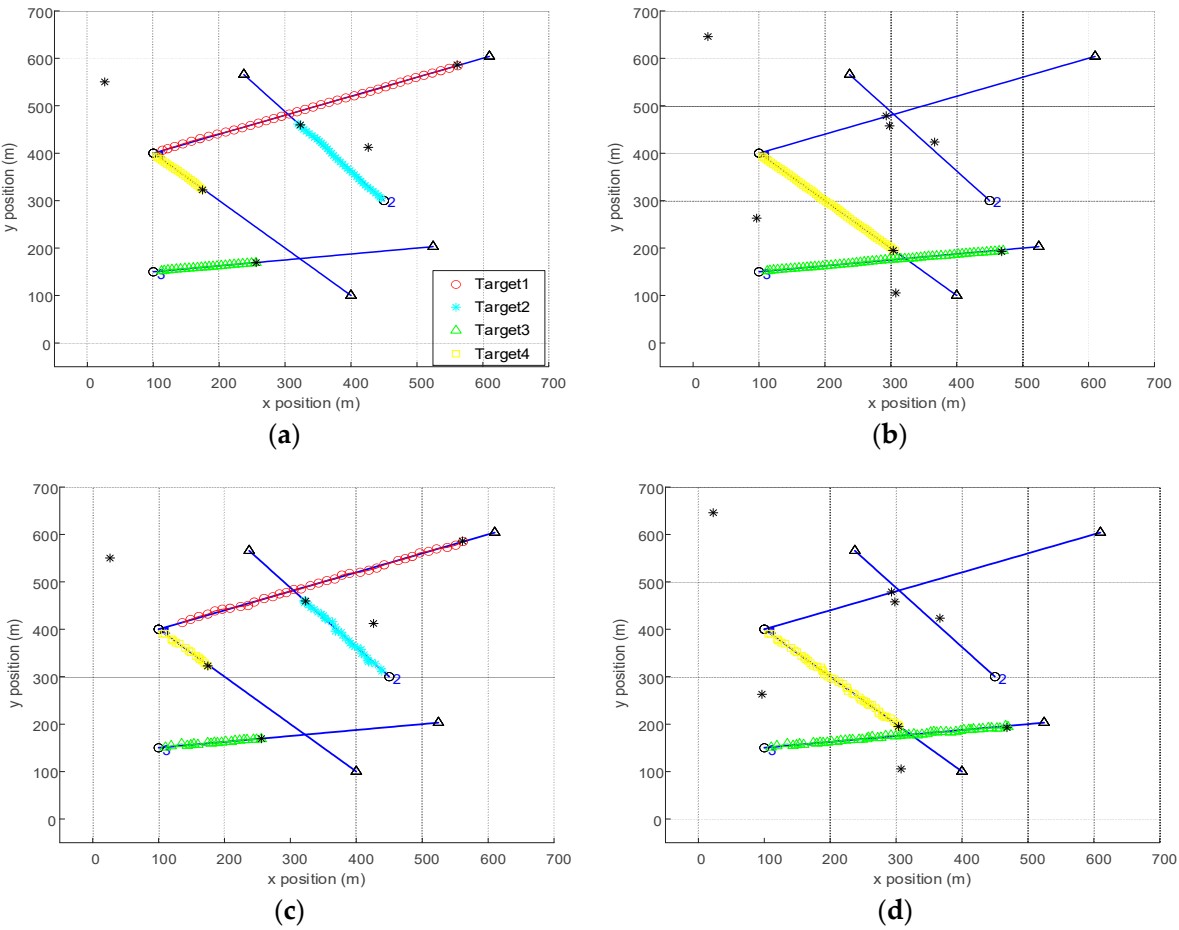

**Figure 1.** (**a**) The trajectory estimates of AGM-VB-TPHD until time 40. (**b**) The trajectory estimates of AGM-VB-TPHD until time 75. (**c**) The trajectory estimates of AGM-tag-VB-PHD until time 40. (**d**) The trajectory estimates of AGM-tag-VB-PHD until time 75. The blue line represents the true trajectories of the targets in the period, the circle represents the starting point of the trajectory, the triangle represents the end point of the trajectory, different trajectories are marked with different color styles and the black asterisk represents the measurement at the current time.

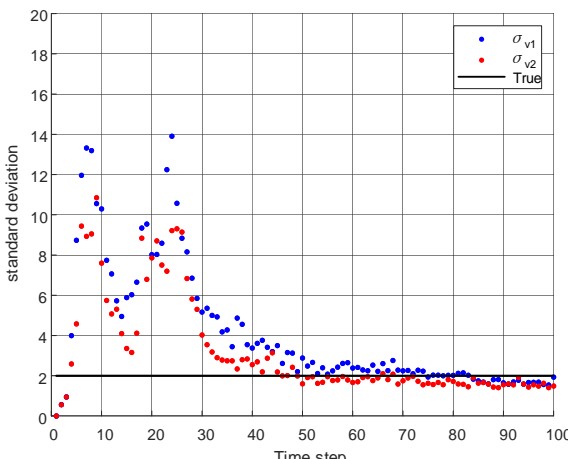

**Figure 2.** The average standard deviation of estimated measurement noise of AGM-VB-TPHD.

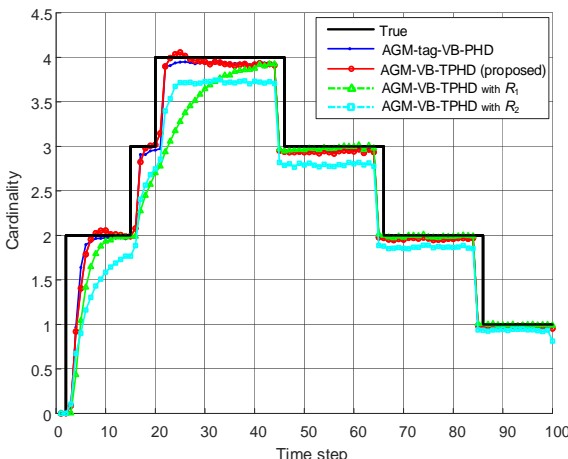

**Figure 3.** The average cardinalities of different filters.

Figure 2 shows the estimation process of AGM-VB-TPHD filter for the unknown measurement noise standard deviation. In the period with new targets' appearance, larger standard deviation estimation may occur. However, as time goes by, the two independent components of standard deviation approach the unknown true standard deviation adaptively and gradually tend to be stable. To highlight the performance of AGM-VB-TPHD filters, we conducted the simulations of the adaptive newborn intensity Gaussian mixture tag-VB-PHD (AGM-tag-VB-PHD) filter [20,34], AGM-TPHD filter with larger estimated covariance ($R_1 = \mathrm{diag}(40, 40)$) and smaller estimated covariance ($R_2 = \mathrm{diag}(0.25, 0.25)$) for comparation under the same parameters.

The effective number of tracking targets at each time can be seen from the cardinality distribution. The closer the cardinality to the true value is, the lower the target losing probability is, and the more stable the tracking performance is. Figure 3 shows that AGM-VB-TPHD filter and AGM-tag-VB-PHD filter have little difference in the estimation of target number and are more accurate than that whose covariance estimation is wrong. When the estimated covariance is $R_1$, the position parameters of the covariance of the new intensity are much smaller than the estimated covariance parameters, resulting in the estimation mismatch, so that the filter cannot detect the target immediately while a new target is generated. When the estimated covariance is $R_2$, it can be seen from Figure 3 and the GOSPA missed target cost in the Figure 4 that the degree of target missed detection is serious. This is because the covariance estimation is too small to cover the range of targets that should be detected, resulting in a large loss of targets.

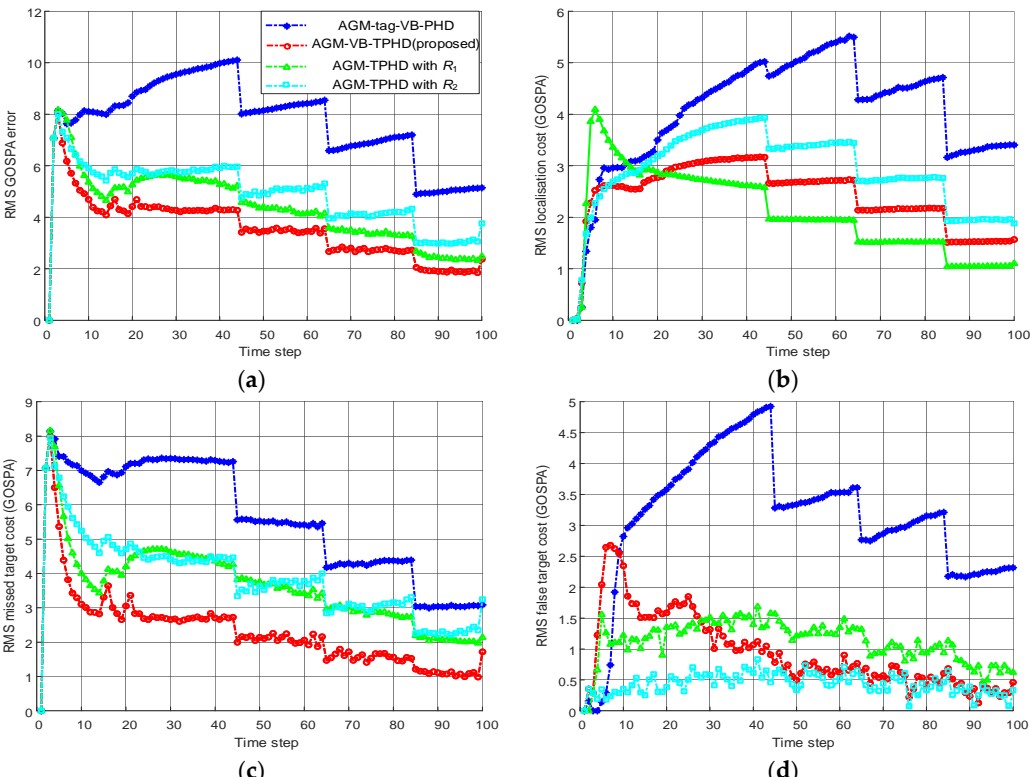

**Figure 4.** Comparison of adaptive and false estimation of AGM-VB-TPHD as well as AGM-tag-VB-PHD under GOSPA metric. (**a**) RMS GOSPA error of filters. (**b**) RMS GOSPA localization cost of filters. (**c**) RMS GOSPA missed target cost of filters. (**d**) RMS GOSPA false target cost of filters.

From the square GOSPA error on the upper left of Figure 4, we can see that the error of AGM-VB-TPHD filter is the smallest and the tracking effect is the best. The performance of the AGM-VB-TPHD filter is obviously better than that of the AGM-tag-VB-PHD filter under the three indexes of localization, missed detections and false targets. The AGM-tag-VB-PHD filter is based on labeling each PHD component. It does not improve the accuracy of the PHD itself, and if multiple components extracted have the same label, it will cause missed target and false detection. A larger estimated covariance will lead to a larger range of target detection. As part of the measurement is redundant clutter, there will be the generation of wrong targets. A large covariance can lead to a small estimated error of target localization, but it is still inferior to the AGM-VB-TPHD filter under the overall GOSPA metric. If the estimated covariance is too small, the target will be missed. After the adaptive covariance is stabilized, the AGM-VB-TPHD filter shows the best performance in the three indicators clearly. Based on overall evaluation, the AGM-VB-TPHD filter has the smallest GOSPA error and the most accurate number of estimated trajectories.

*5.2. Scenario 2*

To prove the tracking performance of VB-TPHD in nonlinear scenarios and the effect of update window length $L$ on filtering, we design a two-dimensional extended trajectory space scenario with unknown and time-varying noise covariance. Suppose there are two targets that can form trajectories in the scenario and each target makes a constant turning motion. The dynamic and measurement model are shown in the Equations (57) and (58):

$$
F_k = \begin{bmatrix} 1 & \frac{\sin(\omega)}{\omega} & 0 & \frac{-(1-\cos(\omega))}{\omega} & 0 \\ 0 & \frac{1-\cos(\omega)}{\omega} & 1 & \frac{\sin(\omega)}{\omega} & 0 \\ 0 & \cos(\omega) & 0 & -\sin(\omega) & 0 \\ 0 & \sin(\omega) & 0 & \cos(\omega) & 0 \\ 0 & 0 & 0 & 0 & 1 \end{bmatrix}, B_k = \begin{bmatrix} \sigma_v \frac{\Delta^2}{2} & 0 & 0 \\ \sigma_v \Delta & 0 & 0 \\ 0 & \sigma_v \frac{\Delta^2}{2} & 0 \\ 0 & \sigma_v \Delta & 0 \\ 0 & 0 & \sigma_t \Delta \end{bmatrix}, \tag{60}
$$

$$Q_k = B_k \times B_k{}^T, \quad h_k(x_k, 0) = \begin{bmatrix} \arctan(\frac{p_{x,k}}{p_{y,k}}) \\ \sqrt{p_{x,k}^2 + p_{y,k}^2} \end{bmatrix}, \quad R_k = \mathrm{diag}(\sigma_{v_r}^2, \sigma_{v_d}^2)$$

where, $\Delta = 1$ s, $\sigma_v = 1.8$ m/s represents the velocity standard deviation of the process noise and $\sigma_t = (\pi/180)^2$ rad/s represents the angular velocity standard deviation of the process noise. $\sigma_{v_r}$ and $\sigma_{v_d}$ represent the unknown parameters of the standard deviation of the measurement noise. The former four dimensions of the state quantity of the target in the extended space are the same as the linear model and the fifth dimension is the angular velocity of the target. The first and second dimensions are the azimuth and distance of the relevant observation point (set as the origin of coordinates). Next are some settings for the scenario parameters. The scenario size is set to $[0, \pi/2]$ rad $\times$ $[0, 1200]$ m and the total tracking duration is set to 100 s. The trajectory survival probability is set as $p_s = 0.99$ and the detection probability is set as $p_D = 0.99$. The clutter generated in each scan obeys the uniform distribution and its quantity obeys the Poisson distribution with parameter $\lambda = 3$.

The initial state, time of birth and time of death and measurement noise covariance of the trajectory in the extended state space are shown in Table 2. Note that target 2 is divided by the bound of time 50 and the standard deviation of measurement noise in the first period is diag$(\pi/180, 2)$ and in the second period is diag$(1.5\pi/180, 3)$. The threshold of pruning is set as $\varepsilon_p = 10^{-4}$, the threshold of absorption is set as $\varepsilon_a = 4$ and the maximum number of Gaussian components is set as $J_{\max} = 70$. The covariance matrix for quantitative measurement of noise obeys the inverse Gamma distribution. The initial parameters are $\alpha_1 = 1$, $\beta_1 = \pi/180 \times 0.01$, $\alpha_2 = \beta_2 = 1$, and the given attenuation factor is $\rho = 0.95$.

**Table 2.** The information of true targets.

| | Initial Targets States | Birth Time/s | Death Time/s | $\sigma_{v_r}$ | $\sigma_{v_d}$ |
|---|---|---|---|---|---|
| Target 1 | $\{100; 550; 6.76; 0.95; -\pi/207\}$ | 2 | 100 | $2\pi/180$ | 5 |
| Target 2 | $\{600; 600; 1.45; 5.99; -\pi/207\}$ | 15 | 100 | $\pi/180; 1.5\pi/180$ | 2;3 |

Similar to the linear scenario simulation, we also conducted 1000 Monte Carlo runs on the trajectory set with MATLAB 2021a on 2.50 GHz Intel I5 laptop. In scenario 2, the trajectory metric [40] is used to evaluate the performance of the extended trajectory estimation, and the parameters are set as $p = 2$, $c = 10$, $\gamma = 1$. The target RMS trajectory metric error (TM) consists of localization, missed targets, false targets and track switching.

Figures 5–9 show when the length $L$ of the update window is set to 5. Figures 5 and 6 respectively show the trajectories of AEK-VB-TPHD filter with the AUK-VB-TPHD filter and the average value after the Monte Carlo runs of standard deviation of measurement noise. It can be seen that even under the condition of nonlinear and time-varying covariance, the VB-TPHD filter can still perform well in the estimation of trajectory and noise standard deviation. In Figure 5, the tag-VB-PHD filter delivers poor performance. When AUK-tag-VB-PHD is used to track target 2, there is a series of wrong targets deviated from the true trajectory, and tremendous trajectory switching errors are produced after targets cross.

In terms of cardinality distribution, the AEK-VB-TPHD filter and AUK-VB-TPHD filter in Figure 7 are significantly improved, whose estimated numbers are closer to the true value compared with the AEK-tag-VB-PHD filter and the AUK-tag-VB-PHD filter. In scenario 2, the trajectory metric [40] is used to evaluate filter performance and the influence of different L-scan window lengths on filter estimation performance is compared. From Figures 8 and 9, it can be seen that the AEK-VB-TPHD filter and the AUK-VB-TPHD filter are basically superior to the AEK-tag-VB-PHD filter and the AUK-tag-VB-PHD filter in the trajectory measurement; additionally, the former two filters are very close under the trajectory metric. AEK-tag-VB-PHD and AUK-tag-VB-PHD provide considerable errors in missing targets, false targets and trajectory switching obviously, especially when two trajectories are close or cross.

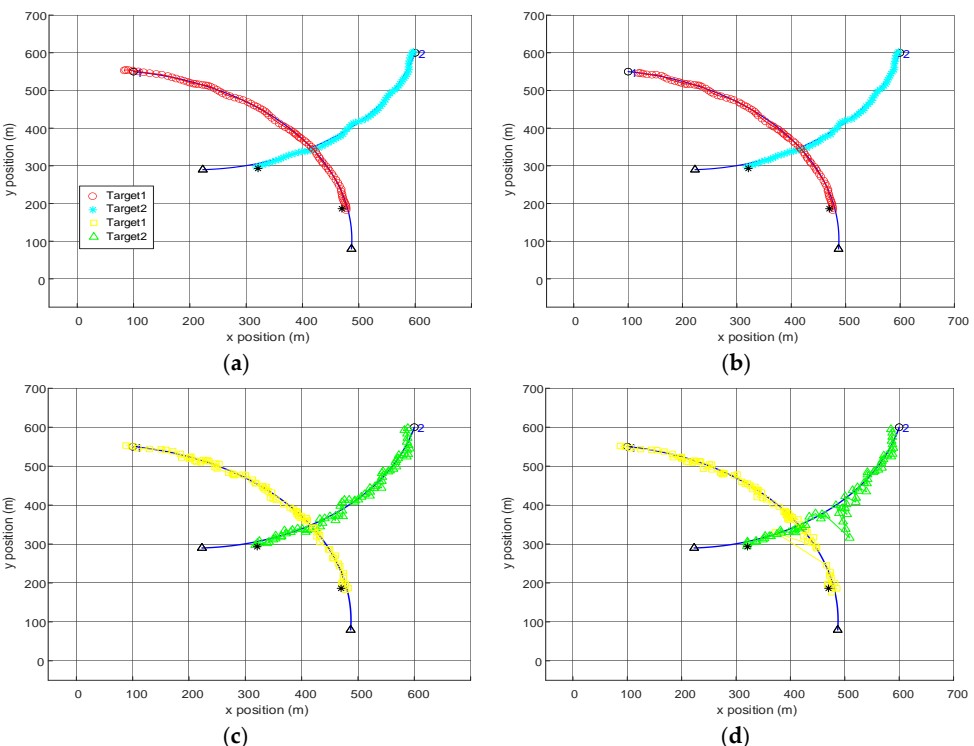

**Figure 5.** (**a**) The trajectory estimates of AEK-VB-TPHD until time 85. (**b**) The trajectory estimates of AUK-VB-TPHD until time 85. (**c**) The trajectory estimates of AEK-tag-VB-PHD until time 85. (**d**) The trajectory estimates of AUK-tag-VB-PHD until time 85. The trajectory of the true target is marked with blue line segments in the period, the circle represents the starting point of the track, the triangle represents the end point of the trajectory, different trajectories are marked with different color styles (*L* = 5) and the black asterisk represents the measurement at the current time.

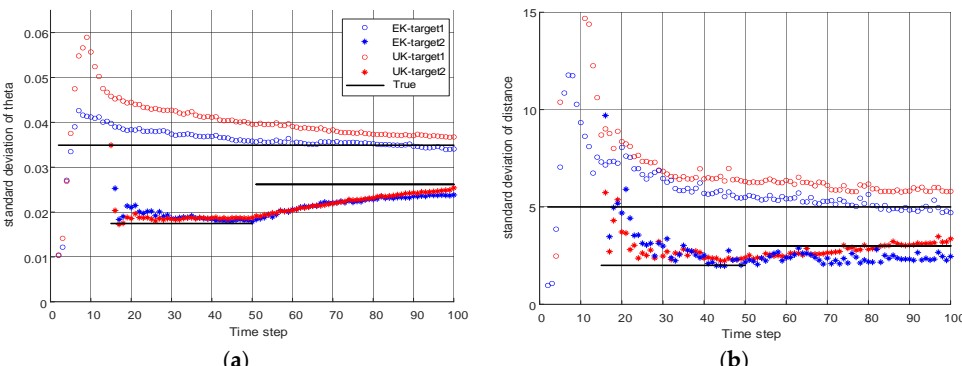

**Figure 6.** (**a**) Standard deviation estimation of theta of AEK-VB-TPHD and AUK-VB-TPHD. (**b**) Standard deviation estimation of distance of AEK-VB-TPHD and AUK-VB-TPHD (*L* = 5).

Table 3 shows the average time required by AEK-VB-TPHD and AUK-VB-TPHD under different window lengths to achieve similar covariance estimation effect. With the increase of *L*, the time of the two filters both increase. The time increases dramatically especially after *L* = 10. On the other hand, the time of AUK-VB-TPHD is basically almost 3 times that of AEK-VB-TPHD. Table 4 combined with Figure 10 shows the RMS TM errors of AUK-VB-TPHD and AEK-VB-TPHD under different filter windows *L*. Increasing *L* can reduce the RMS TM error and improve the estimation performance of trajectory. When *L* is less than 5, increasing *L* can improve the trajectory performance significantly. In addition, the filtering error of AUK-VB-TPHD is slightly smaller than that of AEK-VB-TPHD, which has a slight advantage in filtering performance.

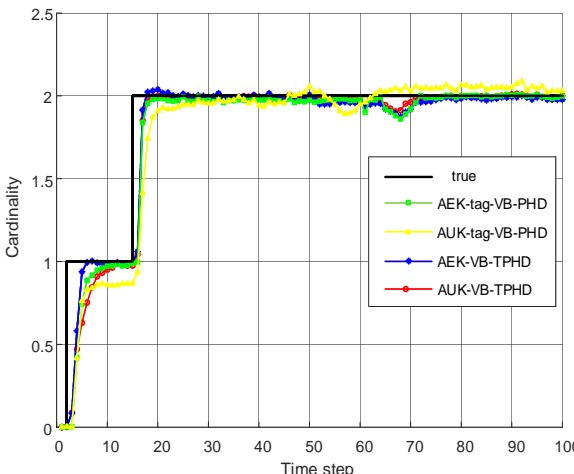

**Figure 7.** The black line represents the cardinality of true targets, the blue line represents the cardinality estimation of AEK-VB-TPHD, the red line represents the cardinality estimation of AUK-VB-TPHD and the green line represents the cardinality estimation of AEK-tag-VB-PHD. The yellow line represents the cardinality estimation of AUK-tag-VB-PHD ($L = 5$).

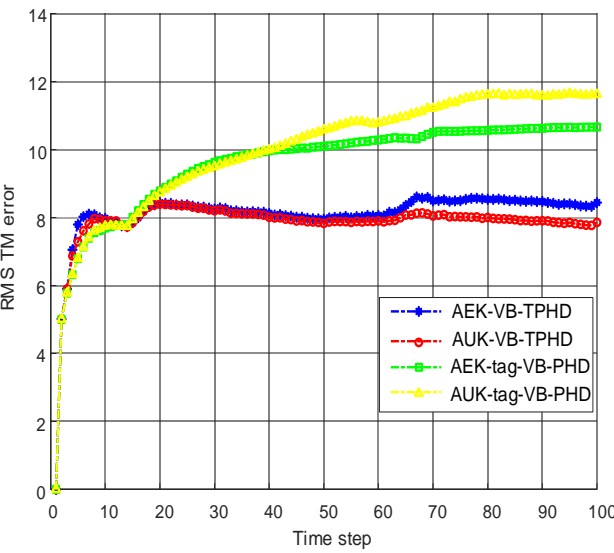

**Figure 8.** Comparison of VB-TPHD and tag-VB-PHD using EK and UK nonlinear realization forms under RMS TM metric under time-varying noise measurement ($L = 5$).

**Table 3.** The average run time of the AEK-VB-TPHD and AUK-VB-TPHD filters.

| L | 1 | 2 | 5 | 10 | 30 |
|---|---|---|---|---|---|
| AEK-VB-TPHD | 1.6515 | 1.8393 | 2.0058 | 2.4287 | 5.8131 |
| AUK-VB-TPHD | 3.0019 | 3.3813 | 4.7062 | 6.9537 | 21.7574 |

**Table 4.** The RMS TM error of the AEK-VB-TPHD and AUK-VB-TPHD filters.

| L | 1 | 2 | 5 | 10 | 30 |
|---|---|---|---|---|---|
| AEK-VB-TPHD | 9.6936 | 9.0836 | 8.1520 | 8.0454 | 7.9827 |
| AUK-VB-TPHD | 9.6791 | 9.0779 | 7.9009 | 7.6279 | 7.5668 |

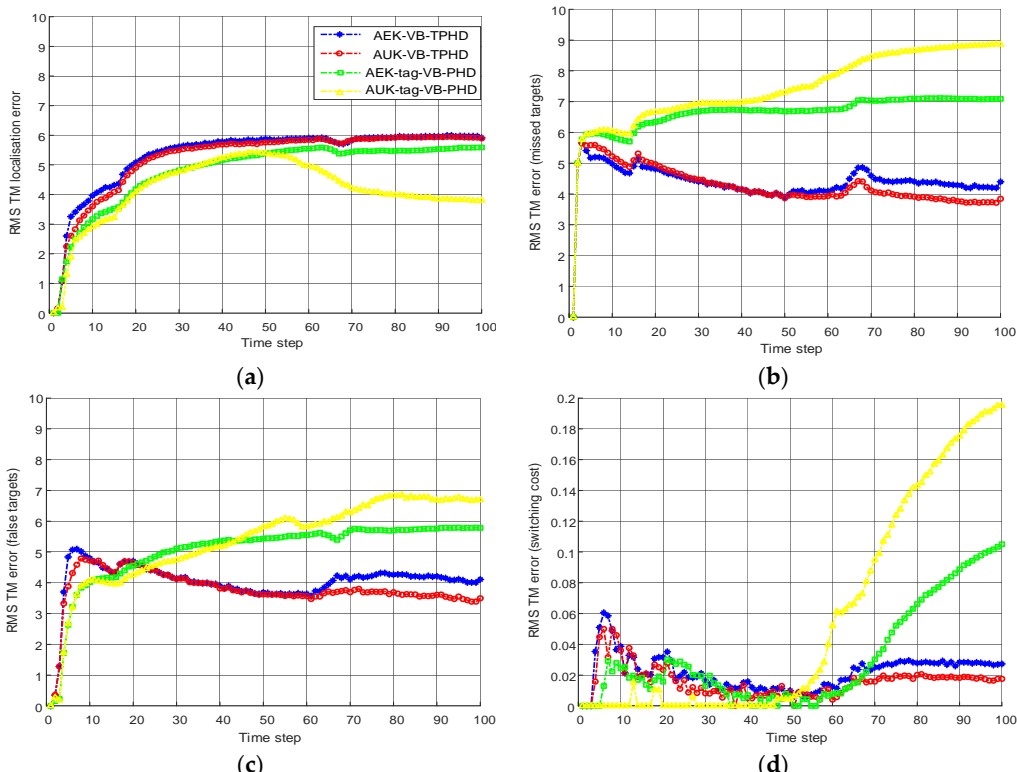

**Figure 9.** Comparison of VB-TPHD and VB-PHD using EK and UK two nonlinear implementation forms under the decomposition of RMS TM metric with time varying measurement noise. (**a**) RMS TM error of localization. (**b**) RMS TM error of false targets. (**c**) RMS TM error of missed targets. (**d**) RMS TM error of switching cost (*L* = 5).

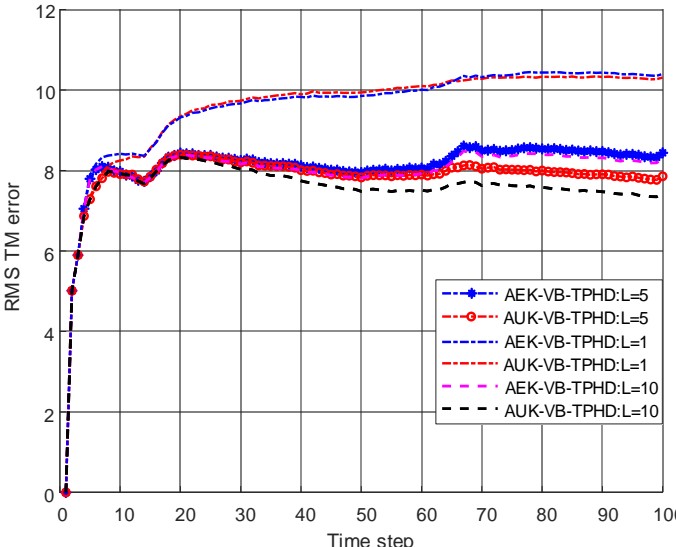

**Figure 10.** Comparison of RMS TM error under different lengths of update window.

## 6. Conclusions

In this paper, we construct the extended trajectory state space under the background of unknown measurement noise covariance, derive the VB-TPHD closed solution by introducing the variational Bayesian approximation into the TPHD filter framework and obtain the joint estimation of the noise covariance sequence and the posterior trajectory state. Considering the filtering efficiency, we only include the noise parameters at the current time in the updating process.

Under the Gaussian model, the extended trajectory state can be modeled as the product of the inverse gamma with the Gaussian mixture. We use the AGM-VB-TPHD filter to complete the trajectory estimation at a constant velocity under the linear model. In addition, AEK-VB-TPHD and AUK-VB-TPHD also have good application effects on the trajectory estimation of the constant angular velocity turns under the nonlinear model. It can be seen from the simulation that VB-TPHD can obtain the statistical information of the dynamic trajectory state with measurement noise effectively and accurately. It improves the estimation of trajectory state and number compared with the adaptive filter based on tag-PHD. The results show the validity and effectiveness of our algorithm along with the implementation and prove that TPHD is an excellent filtering framework.

We suppose that the proposed filter can be applied into maneuvering target detection and the tracking of a harsh and unknown environment in radar and sonar systems. Additionally, the fault-tolerant systems with low-cost integrated GPS/INS positioning systems would be suitable. The next research direction is to apply extended targets and group targets to the framework of TPHD so as to achieve better and more accurate multi-target tracking from the perspective of trajectory.

**Author Contributions:** X.L. performed the simulations and wrote the paper; D.J. (Defu Jiang), Y.G. and D.J. (Dahai Jing) supervised the task and revised the paper with constructive suggestions; J.Y., M.L. and J.T. analyzed the data. Y.L. and W.L. offered revision during the whole process. All authors have read and agreed to the published version of the manuscript.

**Funding:** This research was funded by National Natural Science Foundation of China under grant 61971179 and the Fundamental Research Funds for the Central Universities under grant B200202165.

**Institutional Review Board Statement:** Not applicable.

**Informed Consent Statement:** Not applicable.

**Data Availability Statement:** Not applicable.

**Conflicts of Interest:** The authors declare no conflict of interest.

**Appendix A**

Before proposition 1 is put forward, we discuss the prerequisite of VB-TPHD filter that the measurement noise covariance is stochastic with the independent dynamic models and the survival probability along with detection probability are also independent of the covariance, $p_{s,k}(\overline{X}) = p_{s,k}(x^{i-1})$, $p_{D,k}(\overline{X}) = p_{D,k}(x^i)$. Therefore, according to [29], there is the equation as follows:

$$f\left(x^i, R^i | x^{i-1}, R^{i-1}\right) = f\left(x^i | x^{i-1}\right) g\left(R^i | R^{i-1}\right) \qquad (A1)$$

Connected with Equation (12), the surviving extended trajectory PHD can be written as

$$\begin{aligned} D_{\xi_k}\left(t, x^{1:i}, R^{1:i}\right) &= p_{s,k}(x^{i-1}) f\left(x^i, R^i | x^{i-1}, R^{i-1}\right) D_{\pi_{k-1}}\left(t, x^{1:i-1}, R^{1:i-1}\right) \delta_{N_k}[t] \\ &= p_{s,k}(\overline{X}) f\left(x^i | x^{i-1}\right) g\left(R^i | R^{i-1}\right) D_{\pi_{k-1}}\left(t, x^{1:i-1}, R^{1:i-1}\right) \delta_{N_k}[t], \\ &= p_{s,k}(x^{i-1}) \cdot f\left(x^i | x^{i-1}\right) g\left(R^i | R^{i-1}\right) D_{\pi_{k-1}}\left(t, x^{1:i-1}, R^{1:i-1}\right) \delta_{N_k}[t], \end{aligned} \qquad (A2)$$

where $f(\cdot)$ and $g(\cdot)$ denote state transition density as well as noise covariance transition density. Obviously, the newborn extended trajectory PHD $D_{\gamma_k}\left(t, x^{1:i}, R^{1:i}\right)$ follows by Equation (11):

$$D_{\gamma_k}(\overline{X}) = D_{\gamma_k}\left(t, x^{1:i}, R^{1:i}\right) = D_{\gamma}\left(t, x^{1:i}, R^{1:i}\right) \delta_k[t], \qquad (A3)$$

$$D_{\omega_k}(\overline{X}) = D_{\gamma}\left(t, x^{1:i}, R^{1:i}\right) \delta_k[t] + p_{s,k}(x^{i-1}) \cdot f(x^i | x^{i-1}) g(R^i | R^{i-1}) D_{\pi_{k-1}}\left(t, x^{1:i-1}, R^{1:i-1}\right) \delta_{N_k}[t] \qquad (A4)$$

Therefore, the whole extended trajectory PHD consists of the surviving extended trajectory PHD and the newborn extended trajectory PHD.

**Appendix B**

The update step of single trajectory PHD at time $k$ can be expressed as Equations (13) and (14) in [23]. This theorem shows the target-to-measurement association in multi-trajectory. Now, we want to extend the concept to the extended trajectory space and hope the measuring density can only involve the single target currently instead of the whole trajectory at different times. According to Assumptions 4–6, the density of measurement in the extended trajectory space is referred to by [1,23]

$$
\begin{aligned}
& \ell_k(\{z_1, \ldots, z_n\} | \{\overline{x}_1, \ldots, \overline{x}_m\}) \\
& = e^{-\lambda_c} \left[ \prod_{p=1}^{n} \lambda_c \dot{c}(z_p) \right] \left[ \prod_{p=1}^{m} (1 - p_D(\overline{x}_p)) \right] \\
& \quad \times \sum_{\sigma \in \Xi_{n,m}} \prod_{p:\sigma_p > 0} \frac{p_D(\overline{x}_p) \, l(z_p | \overline{x}_p)}{(1 - p_D(\overline{x}_p)) \lambda_c \dot{c}(z_{\sigma_p})},
\end{aligned}
\tag{A5}
$$

where $\sigma \in \Xi_{n,m}$ represents the whole likely associations between $n$ measurements and $m$ targets whether it can be detected or not, which follows the principle of one measurement for one target. Therefore, we can obtain the pseudolikelihood function of the PHD filter:

$$
L_{Z_k}\left(x^i, R^i\right) = \left(1 - p_{D,k}(x^i)\right) + p_{D,k}(x^i) \times \sum_{z \in Z_k} \frac{l_k(z | x^i, R^i)}{\lambda_c c + \iint p_{D,k}(x^i) l_k(z | x^i, R^i) D_{\omega_k}^\tau(x^i, R^i) dx^i dR^i}
\tag{A6}
$$

$D_{\omega_k}^\tau\left(x^i, R^i\right)$ is given by Equation (21). The final posteriori extended trajectory density can be derived from the pseudolikelihood function of VB-TPHD filter:

$$
\begin{aligned}
D_{\pi_k}(\hat{X}) & = D_{\omega_k}\left(t, x^{1:i}, R^{1:i}\right) L_{Z_k}\left(x^i, R^i\right) = D_{\omega_k}\left(t, x^{1:i}, R^{1:i}\right) \left(1 - p_{D,k}(x^i)\right) \\
& + p_{D,k}(x^i) \times \sum_{z \in Z_k} \frac{l_k\left(z | x^i, R^i\right) D_{\omega_k}\left(t, x^{1:i}, R^{1:i}\right)}{\lambda_c c + \iint p_{D,k}(x^i) l_k\left(z | x^i, R^i\right) D_{\omega_k}^\tau\left(x^i, R^i\right) dx^i dR^i}
\end{aligned}
\tag{A7}
$$

**Appendix C**

Equations (36)–(40) mainly describe the generation of $w_{\xi_k}^j$, $\hat{m}_{\xi_k}^j$, $\hat{P}_{\xi_k}^j$ in the step of prediction under Gaussian model, which can be found in [23]. Hence, it will not be explained too much in this appendix. It is worth noting that $\alpha_{k-1}^{l,j}$ and $\beta_{k-1}^{l,j}$, the freedom and scale parameters of the inverse gamma distribution, increase the covariances by a factor $\rho$ resulting in improving the stability by the VB method through the iterations.

**Appendix D**

The proof of Proposition 4 is described in this appendix. The process of updating in the VB-TPHD filter is on the basis of the TPHD filter, which is elaborated in detail in [23]. Hence, the basic Equations (44)–(50) and (52) will not be explained in this appendix. It is critical to calculate parameters of the inverse gamma distribution estimated with a fixed-point iteration. Section 3.2 obtains posterior distribution of the trajectory state and the noise covariances by a factorized free form distribution. With the concrete distributions in Section 4, $D_{X_k}\left(x_{\pi_k}^{1:i}\right)$ and $D_{R_k}\left(R_{\pi_k}^i\right)$ can be achieved further:

$$
\begin{aligned}
& \int \log D_{D_k}(x_{\pi_k}^{1:i}, R_{\pi_k}^i, z_k | z_{1:k-1}) D_{R_k}(R_{\pi_k}^i) dR_{\pi_k}^i \\
& = -\frac{1}{2}(z_k - \dot{H}\hat{m}_{\pi_k}^j)^T R^{i-1}_R(z_k - \dot{H}\hat{m}_{\pi_k}^j) \\
& \quad -\frac{1}{2}(\hat{m}_{\pi_k}^j - \hat{m}_{\omega_k}^j)^T \hat{P}_{\pi_k}^{j-1}(\hat{m}_{\pi_k}^j - \hat{m}_{\omega_k}^j) + C_1,
\end{aligned}
\tag{A8}
$$

where $\langle \cdot \rangle_R = \int (\cdot) D_{R_k}(R_k) dR_k$ represents the expected value about the approximating distribution of $D_{R_k}(R_k)$ and $C_1$ represents terms independent of $x_{\pi_k}^{1:i}$.

$$
\begin{aligned}
& \int log D_k \left( x_{\pi_k}^{1:i}, R_{\pi_k}^i, z_k | z_{1:k-1} \right) D_{X_k} \left( x_{\pi_k}^{1:i} \right) dx_{\pi_k}^{1:i} \\
& = - \sum_{l=1}^{d} \left( \tfrac{3}{2} + \alpha_{\pi_k}^{l,j} \right) ln \left( \left( \sigma_{\pi_k}^{l,j} \right)^2 \right) - \sum_{l=1}^{d} \frac{\beta_{\pi_k}^{l,j}}{\left( \sigma_{\pi_k}^{l,j} \right)^2} \\
& - \tfrac{1}{2} \sum \frac{1}{\left( \sigma_{\pi_k}^{l,j} \right)^2} \left( z_k - \dot{H} \hat{m}_{\pi_k}^j \right)_l^2{}_X + C_2,
\end{aligned}
\tag{A9}
$$

where $\langle \cdot \rangle_X = \int (\cdot) D_{X_k}(X_k) dX_k$. It can be seen that the parameters of $D_{R_k}(R_k)$ and $D_{X_k}(X_k)$ are able to be found with standard matrix manipulations. We evaluate the expectations in Equations (A8) and (A9) to handle the parameters $\alpha_{\omega_k}^{l,j}$ and $\beta_{\omega_k}^{j,l}$ through this step by matching them with the statistical characteristics of the Gaussian on the left and right sides:

$$
\alpha_{\pi_k}^{l,j} = \alpha_{\omega_k}^{l,j} + 0.5
\tag{A10}
$$

$$
\beta_{\pi_k}^{l,j} = \beta_{\omega_k}^{l,j} + 0.5 \times \left( z - \hat{m}_{\pi_k}^j \right)^2 + 0.5 \times \left( \dot{H} \hat{P}_{\pi_k}^j \left( \dot{H} \right)^T \right)
\tag{A11}
$$

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
