# Peer review of "Trajectory PHD Filter for Adaptive Measurement Noise Covariance Based on Variational Bayesian Approximation"

_applsci, doi:10.3390/app12136388_

Round 1

Reviewer 1 Report

The authors researched an adaptive multi-trajectory algorithm based on the variational Bayesian trajectory probability hypothesis density filter (VB-TPHD filter). The primary filter function is to measure a noise variance.

This is an interesting research report. In my opinion, the manuscript is well-structured, covers an important research topic, and is current. 

I have no major comments, only a few minor comments for authors to address (in terms of applicability):

  1. Please, I advise the authors to consider writing the (experimental) hardware setup at the beginning of Chapter 5 instead of in the line 633. 
  2. What are the potential limits of the algorithm? (in terms of a number of targets, real-time tracking performances, etc.)
  3. How many targets can be tracked with the proposed algorithm with respect to keeping performances (real-time tracking)?

Author Response

Dear reviewer

I am so appreciated your favorable comments on my paper. According to your comments, we revised our paper carefully in the new manuscript entitled as: '' Trajectory PHD Filter for Adaptive Measurement Noise Covariance based on Variational Bayesian approximation''.

Please kindly email us if you have any questions. Million thanks in advance!

Best regards

Mr. Xingchen Lu

Laboratory of Array and Information Processing

Collage of Computer and Information

Hohai University

June 9, 2022

Detailed revisions according to each comment are as follows:

Point 1: Please, I advise the authors to consider writing the (experimental) hardware setup at the beginning of Chapter 5 instead of in the line 633. 

Response 1: Thanks very much for your great suggestion. We carefully reviewed the issue and reset the position of the (experimental) hardware setup. We set up both linear and nonlinear scenarios with the same hardware setup contributing to compare the performance of each filter better and highlight the advantages of VB-TPHD filter. We place the hardware settings after the statement of 1000 Monte Carlo runs instead of the beginning of Chapter 5, due to the same random values of normal distribution with clutter’s position and arithmetic speed in single run of different filters so that the comparation can be carried out. We hope you could understand us on this point sincerely.

Point 2: What are the potential limits of the algorithm? (in terms of a number of targets, real-time tracking performances, etc.)

Response 2: Due to the uncertainty of newborn target location, the prior known intensity is usually artificial to avoid taking the whole detection area into the calculation of target intensity. TPHD filter initializes the newborn intensity from other theoretical literature at the same way. We generate the newborn intensity by the measurements obtained from each scan adaptively and get rid of the dependence of prior known intensity.

Every coin has two sides. In the background of a smaller number of measurements, the proposed filter can achieve ideal results and real-time tracking performances as normal PHD filter. However, the proposed filter cannot handle the situation of tremendous measurements because of calculative burden.

Therefore, there are a few of potential limits of the algorithm. Too many targets or too much clutter resulting to the tremendous measurements can cause the filter processing cannot be carried out and the real-time tracking performances can be greatly affected.

Point 3: How many targets can be tracked with the proposed algorithm with respect to keeping performances (real-time tracking)?

Response 3: According to the experiment we conducted, almost no more than ten targets can be real-time tracked under few clutter. With our limits of hardware setup possibly, the proposed filter cannot handle the environment of tremendous measurements with regard to calculative burden of trajectory instead of targets as usual.

Thanks again for your favorable comments on our paper. Really appreciate!

Reviewer 2 Report

This article is about building an extended state space of the trajectory against an unknown measurement noise covariance, deriving a closed VB-TPHD solution by introducing a variational Bayesian approximation into the TPHD filter structure, and deriving a joint estimate of the noise covariance sequence and a posteriori estimate. trajectory state. Considering the efficiency of filtering, the authors include only noise parameters at the current time in the update process. It can be seen from the simulation that VB-TPHD can effectively and accurately obtain statistical information about the dynamic state of the trajectory as well as noise. Improves state and target count estimation compared to tag-PHD based adaptive filter.

Comments.

1. The abstract should contain more phrases for a wide range of readers. It is necessary to define the scope of the results in understandable language.

2. It seems to me that the designation 1:i is not generally accepted. You need to add the notation used.

3. It would be nice if the Discussion section here would form a discussion of the results.

In general, the article can be accepted.

Author Response

Dear reviewer

I am so appreciated your favorable comments on my paper. According to your comments, we revised our paper carefully in the new manuscript entitled as: ''Trajectory PHD Filter for Adaptive Measurement Noise Covariance based on Variational Bayesian approximation''.

Please kindly email us if you have any questions. Million thanks in advance!

Best regards

Mr.  Xingchen Lu

Laboratory of Array and Information Processing

Collage of Computer and Information

Hohai University

June 8, 2022

Detailed revisions according to each comment are as follows:

Point 1: The abstract should contain more phrases for a wide range of readers. It is necessary to define the scope of the results in understandable language.

Response 1: Thanks very much for your great suggestion. We carefully reviewed the issue and rewrite the abstract based on these five key words: trajectory PHD filter; Variational Bayesian approximation; noise covariance matrices; inverse Gamma distribution; estimation of trajectory. We delete the intricate phrases and recombine the sentences carefully.

Abstract: In order to solve the problem that the measurement noise covariance may be unknown or change with time in actual multi-target tracking, this paper brings the Variational Bayesian approximation method into trajectory probability hypothesis density (TPHD) filter and proposes a Variational Bayesian TPHD (VB-TPHD) filter to obtain measurement noise covariance adaptively. By modeling the unknown covariance as the random matrix that obeys the inverse Gamma distribution, VB-TPHD filter minimizes Kullback-Leibler Divergence (KLD) and estimates the sequence of multi-trajectory states with noise variance matrices simultaneously. We propose Gaussian mixture VB-TPHD (AGM-VB-TPHD) filter under adaptive newborn intensity for linear Gaussian models and also give extended Kalman (AEK-VB-TPHD) filter and unscented Kalman (AUK-VB-TPHD) filter in nonlinear Gaussian models. The simulation results prove the effectiveness of the idea that VB-TPHD filter can form robust and stable trajectory filtering while learning adaptive measurement noise statistics. Compared with tag-VB-PHD filter, the estimated error of VB-TPHD filter is greatly reduced and the estimation of trajectory number is more accurate.

Point 2: It seems to me that the designation 1:i is not generally accepted. You need to add the notation used.

Response 2: In our paper, we use the designation 1:i to represent the trajectory duration and the single trajectory can be expressed as the basic variable , according to Ref. [23] García-Fernández, Á.F.; Svensson, L. Trajectory PHD and CPHD Filters. IEEE Trans. Signal Process. 2019, 67, 5702-5714. 

Please see the attachment. This is a picture about the definition of trajectories from the original literature. I also think that the expression 1:i is not generally accepted however it's more accurate and appropriate to use the designation 1:i according to the original literature. We hope you will understand our reason for this action.

Point 3: It would be nice if the Discussion section here would form a discussion of the results.

Response 3: Thank you very much for your favorable considerations on our paper and give the good comment to improve our paper. According to your comments, we carefully reviewed discussion of the results and agreed that we need to add more details to convince the proposed method. Simulation results and discussion is hard to split, so that we combine the two thing into Section 5. The cardinality and RMS GOSPA are the evaluation indexs of the simulation. We believe it would be proper to show the results along with discussion contributing to a better reading experience for readers.

We added subsection 5.1 Page 16 and 17 as follows:

The effective number of tracking targets at each time can be seen from the cardinality distribution. The closer the cardinality to the true value is, the lower the target losing probability is and the more stable the tracking performance is. Figure 3 shows that AGM-VB-TPHD filter and AGM-tag-VB-PHD filter have a little difference in the estimation of target number and are more accurate than that whose covariance estimation is wrong. When the estimated covariance is R1, the position parameters of the covariance of the new intensity are much smaller than the estimated covariance parameters, resulting in the estimation mismatch, so that the filter cannot detect the target immediately while a new target is generated. When the estimated covariance is R2, it can be seen from Figure 3 and the GOSPA missed target cost in the Figure 4 that the degree of target missed detection is serious. This is because the covariance estimation is too small to cover the range of targets that should be detected, resulting in a large loss of targets. 

Figure 3. The average cardinality of different filters.

From the square GOSPA error on the upper left of Figure 4, we can see that the error of AGM-VB-TPHD filter is the smallest and the tracking effect is the best. The performance of AGM-VB-TPHD filter is obviously better than that of AGM-tag-VB-PHD filter under the three indexes of localisation, missed detections and false targets. The AGM-tag-VB-PHD filter is based on labeling each PHD component. It does not improve the accuracy of PHD itself and if multiple components extracted have the same label, it will cause missed target and false detection. A larger covariance estimate will lead to a larger range of target detection. As part of the measurement is redundant clutter, there will be the generation of wrong targets. A large covariance can lead to a small estimation error of target localisation, but it is still inferior to AGM-VB-TPHD filter under the overall GOSPA metric. If the estimated covariance is too small, the target will be missed. After the adaptive covariance is stabilized, the AGM-VB-TPHD filter shows the best performance in the three indicators clearly. Based on overall evaluation, AGM-VB-TPHD filter has the smallest GOSPA error and the most accurate trajectory number of estimation.

Figure 4. Comparison of adaptive and false estimation of AGM-VB-TPHD as well as AGM-tag-VB-PHD under GOSPA metric. (a) RMS GOSPA error of filters. (b) RMS GOSPA localisation cost of filters. (c) RMS GOSPA missed target cost of filters. (d) RMS GOSPA false target cost of filters.

Thanks again for your favorable comments on our paper. Really appreciate!

Reviewer 3 Report

Dear Editor,

The new referee report you requested is attached containing the questions and answers you want,

Respects.

New Review Report(mdpi- applsci –AS-Manuscript ID Number-applsci-1761531)-mcancan

Article title:         Trajectory PHD Filter for Adaptive Measurement Noise Covariance based on Variational Bayesian approximation

Possible questions and answers about the article:

1. What is the main question addressed by the research?

                In the article, the illustrates, necessary properties, definitions and theorems,formulas in finding variationality features of a special “Variational Bayesian equation” structure modelling are characterized different type ideal equations(based (TPHD) filter structure ideas and structure special rules).

2. Do you consider the topic original or relevant to the field? Does it address a specific gap in the field?

                This topic fills the specific gap in characterizing the original “physical algebraic,topological geometrian and Application physical maps , applied sciences, representation approach, comparative analysis, fluid theory, numerical theory” model issue in applied science fields (with mechanical algebra theory and subsets of new type measurement noise covariance sets).

3. What does it add to the subject area compared with other published material?

                The compared with other published material,there are adding “in determining special idea approach field, sufficient conditions for a special complex trajectory PHD filter equations and its contents to be the modelling with special filter set structures on the filter approach theory” the subject area(under trajectory PHD filtersurface structures).

4. What specific improvements should the authors consider regarding the methodology? What further controls should be considered?

                Authors can improve by developing new aspects of a special absorbing pressure the inverse Gamma distribution approach structure on connecting with the structure Kullback-Leibler Divergence (KLD)-problem equation to the associative algorithm representation theory equations of common the best Poisson approximation balancing problem transforms regarding the Methodology, The authors can provide a control mechanism by making new different aspects from the phenomenon of developing special an adaptive newborn intensity Gaussian mixture Variational Bayesian TPHD (AGM-VB-TPHD) filter approach features(in frequency extended Kalman (AEK-VB-TPHD) and unscented Kalman (AUK-VB-TPHD) in nonlinear Gaussian models. approximation method models).

5. Are the conclusions consistent with the evidence and arguments presented and do they address the main question posed?

                The results appear consistent with the evidence and arguments.

The arguments presented adequately address the main question.

6. Are the references appropriate?

                In the article, references appear to be relevant to the study subject.

7. Please (if any) include any additional comments on the tables and figures.

                model mechanisms formed from special applied sciences axiomity features on physical estimation of trajectory approach theory and special idea counting physical -algebraic-mapping forms are developed and shown in terms of article images (under the structure idea connections and with nonlinear Gaussian models).

report:In the article, it is seen that  the special " in physical -algebraic structure,and applied sciences field method(the theory of applied science)( applied theoretic method)" featured definitions and calculations are made (the special mapping forms and enumerating the numbers of physical algebraic on a special net) for some special applied approach form equations (with idea representation and strong enumerate the number calculations under Variational Bayesiang platform, an adaptive newborn intensity Gaussian mixture Variational Bayesian TPHD (AGM-VB-TPHD) filter).

findings: In the article, the special “applied sci and structural properties of the common complex ideas” process approach form equations are seen (the special ideal calculations) with the help of some special forms (under special idea physical algebraic map and ideality -theoretic tool and with a special operation).

strengths: In the article, there are proficiency-enhancing conditions for the special applied idea approach forms.

weaknesses: In the article, it may be necessary to develop some extra new ideal flow balancing conditions for the special ideal filter process algorithms and applied idea approach forms of this type(with axiomatic ideas and mechanisms).

any minor issue: Although there are some minor weaknesses in terms of structure language, the article can be considered sufficient.

result: The article is suitable for publication for the journal.

Author Response

Dear reviewer

I am so appreciated your favorable comments on my paper. According to your comments, we revised our paper carefully in the new manuscript entitled as: '' Trajectory PHD Filter for Adaptive Measurement Noise Covariance based on Variational Bayesian approximation''.

Please kindly email us if you have any questions. Million thanks in advance!

Best regards

Mr. Xingchen Lu

Laboratory of Array and Information Processing

Collage of Computer and Information

Hohai University

June 8, 2022

Detailed revisions according to each comment are as follows:

Point 1: English language and style
(x) Moderate English changes required

Response 1: Thank you very much for your favorable considerations on our paper. And give the good comment to improve our paper. According to your comments, we carefully reedited the grammar and vocabulary of the whole article and moderate English changes as large as possible. We agreed that we need to add more details to convince the proposed method with more concise and accurate English language. 

(Please see the new manuscript)

Point 2: Does the introduction provide sufficient background and include all relevant references?  Can be improved.

Response 2: Thanks very much for your great suggestion. We carefully reviewed the issue and check the introduction once again. Para 1 tells the situation of Multi-target tracking. Para 2 introduces Bayesian filter based on random finite set especially PHD filter. We give the TPHD filter for trajectory in Para 3. Para 4 tells the problem of the covariance and we give our algorithm and implementation in Para 5. We include all references excluded Ref.[38],[39],[40] because their contents are suitable for simulation evaluation indexs or SVD.

(Please see the new manuscript)

Point 3: Are the conclusions supported by the results?  Can be improved.

Response 3: The simulation results show the effectiveness of the idea that VB-TPHD can form robust and stable trajectory filtering while learning measurement noise statistics adaptively on matter in linear or nonlinear models. Compared with tag-VB-PHD filter, the estimation error of VB-TPHD filter is greatly reduced and the estimation of trajectory number is more accurate. Of course, we draw the conclusion that it is correct to introduce the Variational Bayesian approximation into the TPHD filter framework and we achieve the joint estimation of the noise covariance sequence and the posterior trajectory state we want.

(Please see the new manuscript)

Point 4: weaknesses: In the article, it may be necessary to develop some extra new ideal flow balancing conditions for the special ideal filter process algorithms and applied idea approach forms of this type(with axiomatic ideas and mechanisms).

We construct the extended trajectory state space under the background of unknown measurement noise covariance, derive the VB-TPHD closed solution by introducing the Variational Bayesian approximation into the TPHD filter framework and obtain the joint estimation of the noise covariance sequence and the posterior trajectory state. We hope that we can connect the sequence of unknown measurement noise covariance with multi-trajectory by iterated filtering with special ideal filter algorithms and applied this ideal approach to the new area.

(Please see the new manuscript)

Thanks again for your favorable comments on our paper. Really appreciate!

Reviewer 4 Report

The paper corresponds to the aims and scope of the journal "Applied Sciences". The manuscript is well written, but a number of issues require clarification.

There are a few comments:

1. In formula (1), it is necessary to replace the first symbol after the equal sign. At least it should be larger.

2. In formulas (6) and (7), it is better to use the double integral symbol.

3. The choice of the inverse gamma distribution in line 150 must be convincingly justified. Why, for example, do you not use a much wider family: the generalized gamma distributions?

4. The choice of the normal distributions also requires motivation and explanation.

5. The authors need to add clarifications regarding the proofs of propositions 1-4.

6. Section 4 deals, in fact, with finite normal mixtures. Is it possible to use the continuous normal mixtures in this problem? Or is it not required?

7. In lines 437 and 438, the heading "Table" should be replaced by "Algorithm". I also suggest the authors improve the presentation of the material in these algorithms. You should try to avoid duplication of formulas already encountered in the text, or at least give references to them.

8. Authors should add 1-2 paragraphs to section 6 with a discussion of applications in which their results will be most in demand.

Round 2

Reviewer 4 Report

All responses are clear. The manuscript can be accepted in its current form.